# The role of extracellular vesicle fusion with target cells in triggering systemic inflammation

Praveen Papareddy [1] ✉, Ines Tapken [1,2,3], Keshia Kroh [1,4], Ravi Kiran Varma Bhongir [1], Milladur Rahman [5], Maria Baumgarten[1], Eda Irem Cim[1], Lilla Györffy[1], Emanuel Smeds [1], Ariane Neumann [1], Srinivas Veerla [6], Jon Olinder[7], Henrik Thorlacus[5], Cecilia Ryden[7], Eva Bartakova[8], Michal Holub [8] & Heiko Herwald [1] ✉

Extracellular vesicles (EVs) play a crucial role in intercellular communication by transferring bioactive molecules from donor to recipient cells. As a result, EV fusion leads to the modulation of cellular functions and has an impact on both physiological and pathological processes in the recipient cell. This study explores the impact of EV fusion on cellular responses to inflammatory signaling. Our findings reveal that fusion renders non-responsive cells susceptible to inflammatory signaling, as evidenced by increased NF-κB activation and the release of inflammatory mediators. Syntaxin-binding protein 1 is essential for the merge and activation of intracellular signaling. Subsequent analysis show that EVs transfer their functionally active receptors to target cells, making them prone to an otherwise unresponsive state. EVs in complex with their agonist, require no further stimulation of the target cells to trigger mobilization of NF-κB. While receptor antagonists were unable to inhibit NF-κB activation, blocking of the fusion between EVs and their target cells with heparin mitigated inflammation in mice challenged with EVs.

Over two thousand years ago, the Roman encyclopedist Aulus Cornelius Celsus described for the first time the four cardinal signs of inflammation, i.e. calor (warmth), dolor (pain), tumor (swelling) and rubor (redness)[1]. Since then, research has increased our understanding of the molecular mechanisms behind the induction of inflammatory processes in humans. Today it is generally accepted that the rapid activation of the innate immune system and overwhelming inflammatory responses, known as cytokine storms, are two hallmarks of many inflammatory diseases[2,3]. These reactions can be evoked by damage-associated molecular patterns (DAMPs) and pathogen-associated molecular patterns (PAMPs), leading to the mobilization of NF-κB from the cytosol to the nucleus, where the transcription of proteins such as inflammatory mediators and adhesion molecules is initiated[4].

One important player in the induction of inflammatory reactions are extracellular vesicles (EVs.) They are released for instance in patients with underlying inflammatory complications such as polytrauma and sepsis, and can thereby contribute to life-threatening conditions[5,6]. In vitro studies have shown that EVs are secreted via an exocytotic budding process in which phosphatidylserine translocates

[1]Division of Infection Medicine, Department of Clinical Sciences, Lund University, Lund, Sweden. [2]SMATHERIA gGmbH – Non-Profit Biomedical Research Institute, Hannover, Germany. [3]Center for Systems Neuroscience (ZSN), Hannover, Germany. [4]Department of Viroscience, Erasmus Medical Center, Rotterdam, the Netherlands. [5]Section of Surgery, Department of Clinical Sciences, Lund University, Malmö, Sweden. [6]Division of Oncology and Pathology, Lund, Department of Clinical Sciences, Lund University, Lund, Sweden. [7]Division of Infection Medicine, Helsingborg Hospital and Department of Clinical Sciences Helsingborg, Lund University, Lund, Sweden. [8]Department of Infectious Diseases, First Faculty of Medicine, Charles University and Military University Hospital Prague, Praha, Czech Republic. ✉e-mail: praveen.papareddy@med.lu.se; heiko.hewald@med.lu.se

from the inner to the outer leaflet of the cell membrane[7]. EVs have a diameter ranging from 0.1 to 2 μm and retain the same cell-surface protein pattern as found on the cells of origin. The cytosolic components of EVs contain proteins, lipids, nucleic acids (including DNA, mRNA, and non-coding RNA), and small metabolites[8]. Because of the wide range of bioactive proteins, adhesion molecules, and membrane-anchored receptors, EVs are primed for customized crosstalk with their environment. By fusing with target cells, EVs can transfer their parent cytosolic content and are able to translocate cell membrane attached- and spanning-proteins[9].

The fusion process requires the formation of a SNARE-complex[10] and can be blocked by heparin[11]. SNARE-mediated intracellular fusion involves apart from the SNARE-complex also the Syntaxin binding protein 1 (STXBP1), which is a member of the Sec1/Munc18 family. Although the exact molecular mechanism for fusion is still not completely understood, several studies have suggested that STXBP1 initially binds to SNARE subunit syntaxin-1, to form a heterodimer on the target membrane. Once the secretory vesicle or microparticle comes into close apposition with the target membrane, STXBP1 that is bound to the syntaxin-1 can assemble with VAMP2/synaptobrevin on the membrane of the EV. This pins the two membranes together and exerts the force required for a fusion-pore to open between the two membranes[12,13]. With this mechanism EVs can trigger an activation of their target cells, cause phenotypic modifications, and reprogram cell functions[14]. As EVs are released from the cell of origin into the circulatory system, fusion with other cells can occur at distant sites, contributing to the induction of systemic responses[14].

Evidence is accumulating that EVs can make target cells more susceptible to inflammatory signals. However, the molecular mechanisms underlying the transformation of non-immune cells into cells capable of triggering inflammatory signaling pathways require further investigation. In this study, we therefore aim to elucidate how EVs induce systemic immune responses. We conducted gain-of-function and loss-of-function experiments and found that EVs can make non-responsive cells susceptible to inflammatory agonists. Our in vivo experiments showed that soluble receptor antagonists are unable to dampen the inflammatory activity of EVs, which could explain the limited success of some clinical trials. Finally, we used proteomics and systems biology approaches to identify a panel of signaling proteins that are present in EVs from patients with polytrauma and sepsis, two of the most common and severe systemic inflammatory conditions.

## Results

### EVs released from activated cell carry an inflammatory cargo

The aim of the first set of experiments was to investigate the inflammatory state of extracellular vesicles (EVs) that were released from activated immune cells. A special focus was placed on platelet-derived EVs as they are involved in intercellular signaling under both physiological and pathological conditions[15]. To this end we investigated the fusion of platelet-derived EVs with neutrophils and monocytes under inflammatory ex vivo conditions. Human blood was treated with lipopolysaccharide (LPS) or left unstimulated. Using FACS analysis with antibodies against the surface markers CD42b (platelet-derived EVs), CD66b (neutrophils), CD14 (monocytes) we found that the number of CD42b-positive neutrophils and monocytes significantly increased in LPS-treated human blood compared to control (Fig. 1A). Similar findings were also reported by Chimen et al.[16]. Having shown that EVs fuse with immune cells under ex vivo conditions, we switched to an in vitro model by employing THP-1 cells, a human leukemia monocytic cell line. The ability of THP-1 cells-derived EVs to trigger inflammatory reactions, was studied as illustrated in Fig. 1B. EVs, in the following referred to as LPS-EVs, were isolated from THP-1 cells that were stimulated overnight with lipopolysaccharide (LPS). Thereafter, LPS-EVs were lysed, and subjected to NF-κB Pathway Array analysis,

followed by densitometric analysis of the protein patterns. For comparison, THP-1 cells treated with LPS were lysed and also assessed by an NF-κB Pathway Array analysis. The results show that the levels of TNFRSF3 (tumor necrosis factor receptor superfamily member 3) and TRAIL R1 (tumor necrosis factor-related apoptosis-inducing ligand receptor 1) were greatly increased in LPS-EVs compared to the THP-1 cell lysate control (Fig. 1C) which suggests an important role for TNFα in EV-induced inflammatory responses.

For further analyses, the composition of LPS-EVs and levels of other inflammatory mediators were determined. Lysed LPS-EVs served as control. Figure 1D shows that inflammatory mediators, including IL-1β, IL-1ra, GROα, GM-CSF, VEGF, HGF, TNFα, IL-6, IL-22, and IL-9 were enriched in lysed LPS-EVs, while the concentrations of IL-8 and MCP-1 were higher in the unlysed LPS-EVs compared to lysed LPS-EVs. Together, these experiments demonstrate that EVs are equipped with inflammatory proteins involved in activating NF-κB-dependent pathways.

NF-κB signaling is a complex process that is tightly regulated by the phosphorylation of a number of intracellular signaling proteins[17]. To test whether EV-derived proteins are phosphorylated upon release from LPS-stimulated THP-1cells we used Pro-Q Diamond dye, which selectively stains phosphorylated residues in proteins. THP-1 cells were treated with LPS in a time dependent manner. Cells and collected LPS-EVs were then lysed and incubated with the Pro-Q Diamond dye. Figure 1E (*left panel*) depicts a representative SDS-polyacrylamide gel showing that phosphorylated proteins can be found in the lysed cell and LPS-EV fractions. Subsequent treatment with *Gel Code Blue Safe Protein Stain* was applied to visualize the whole protein content in these samples (Fig. 1E; *right panel*). The findings were further supported by Western blot experiments with an anti-phosphotyrosine antibody, showing that phosphorylation pattern of lysates of LPS-stimulated THP-1 cells differs from that of LPS-EVs (Fig. 1F).

Next we investigated whether proteins involved in interleukin signaling pathway, are among the phosphorylated proteins seen in the LPS-EV fraction. We focused on IL1R1, a MyD88- dependent receptor for IL-1α, IL-1β, and the IL-1ra. By using an immunoprecipitation approach, IL-1R1-complexes were pulled down, separated on SDS PAGE, and probed in Western blot experiments with pIRAK4 (phosphorylated IRAK4) and MyD88 antibodies. As shown in Fig. 1G, both proteins were found in complex with IL1R1 when using immunoprecipitates from THP-1 cells and LPS-EVs. These findings suggest that EVs are equipped with the necessary adaptor proteins for mediating intracellular signaling. Additional Western blot experiments demonstrate that apart from MyD88 and IRAK4, also TRAF2 and TRADD (both proteins are involved in TNFα signaling) became phosphorylated in a time-dependent manner in THP-1 cells and EVs upon stimulation with LPS (Fig. 1H). Together these experiments show that EVs are equipped with inflammatory receptors and phosphorylated signaling proteins that required for an activation of the NF-κB pathway.

### EV-cell fusion with target cells requires the SNARE complex

The efficient translocation of EV-bound receptors and EV-stored intracellular signaling proteins relies on their fusion with target cells. This process is facilitated by SNARE complexes which promote the formation of a tight and stable complex between the vesicle and target membrane. SNARE complexes can be found on cellular membranes as illustrated in Fig. 2A. SNARE proteins facilitate membrane fusion by binding to specific SNAREs on acceptor membranes. This in turn leads to conformational changes that are necessary to maintain membrane contact. The assembly and disassembly of SNARE complexes are regulated by accessory proteins, including Sec1/Munc18-related proteins, synaptotagmins, and complexins[18]. One of these proteins Munc18-1 is encoded by the *STXBP1* gene.

To study the role of Munc18-1 (in the following referred to as STXBP1) in EV-triggered fusion processes, we switched to RAW-Blue™

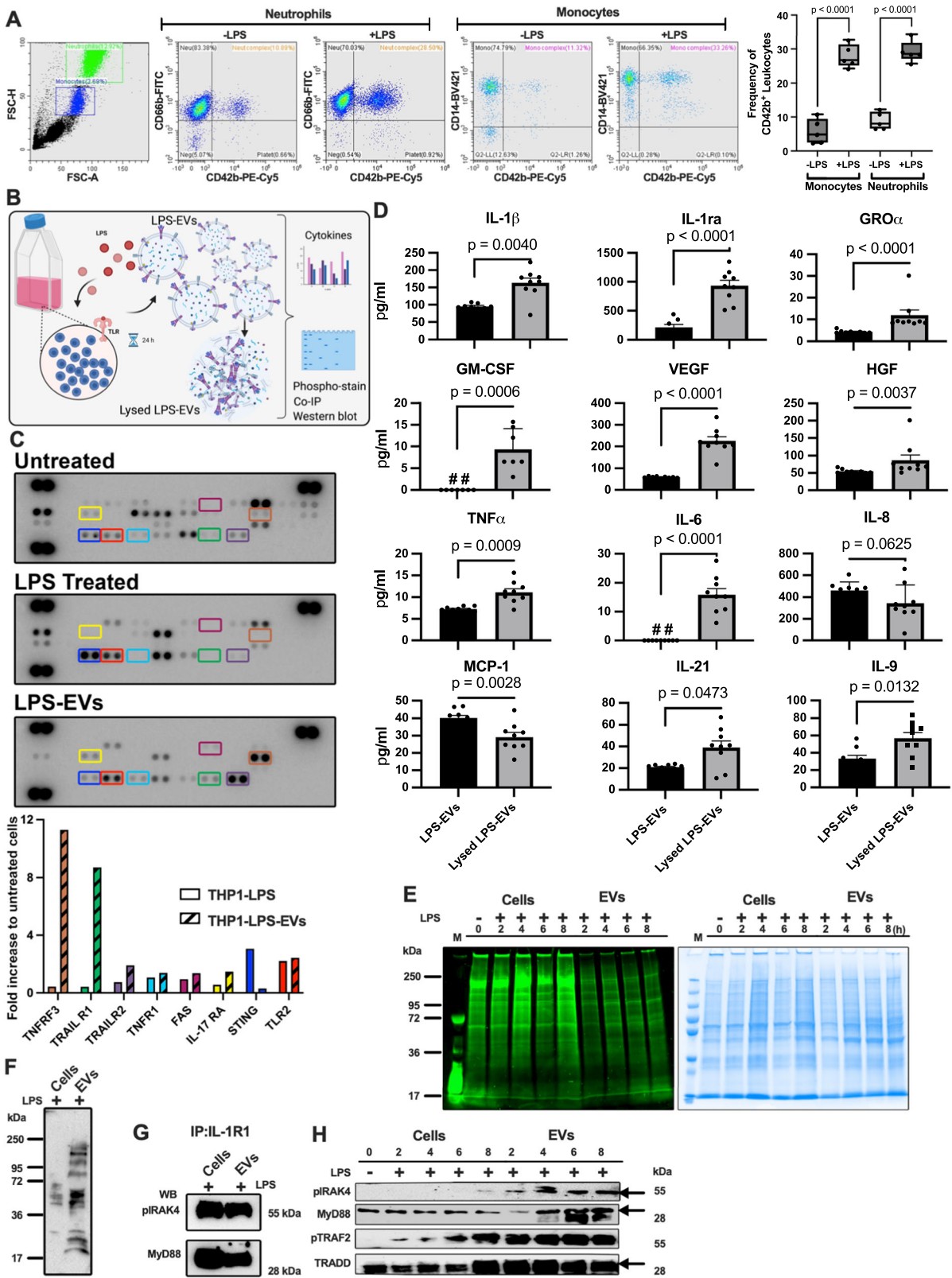

cells for further analyses. RAW-Blue™ cells are a genetically modified murine macrophage-like cell line that expresses an NF-κB/AP-1-inducible SEAP reporter gene. This modification allows monitoring NF-kB and AP-1 responses upon stimulation with inflammatory ligands. To investigate the role of the SNARE complex in promoting fusion of EVs in these cells, the *Stxbp1* gene was additionally silenced by treating RAW-Blue™ cells with specific siRNA. Fig. 2B, C shows that the

treatment resulted in a decrease in both *Stxbp1* mRNA expression and Stxbp1 protein production in RAW-Blue™ cells, as evaluated by RT-PCR and Western blot analysis, respectively. We next tested whether *Stxbp1*-silencing affects RAW-Blue™ cells to respond to stimulation with LPS-EVs. Figure 2D shows that LPS-EVs induce a mobilization of NF-kB in RAW-Blue™ cells, which was decreased when the cells were treated with *Stxbp1* mRNA. When using control siRNA the

**Fig. 1 | EV fusion triggers inflammatory reactions. A** Human blood *(Healthy donors: n = 5)* subjected to flow cytometry analysis post LPS-challenge are presented *(left panel)*. The translocation efficiency of CD42b, a platelet activation marker, to blood cells from healthy and LPS-treated blood is depicted *(right panel)*. Statistical analysis utilized ordinary one-way ANOVA with Tukey's multiple comparisons test, results presented as box-and-whisker plots from minimum value and up to the maximum value, The error bars represent the standard error of the mean (SEM). **B** The schematic diagram illustrates the quantification of EV outer bound proteins and inner packed proteins, including their phosphorylation pattern (Created with BioRender.com). **C** THP-1 cells, stimulated with LPS (10 μg/ml) or PBS, underwent multi-cytokine membrane array analysis *(upper panel)*. Mean pixel densities, calculated via image software, identified up-regulated cytokines (*n = 1*, measurement) denoted in a bar graph *(lower panel)*. **D** LPS-stimulated THP-1 cells were subjected to multiplex immunoassay before and after EV lysis, revealing changes in selected inflammatory mediators. Statistical analysis employed ordinary one-way ANOVA unpaired t-test with Mann-Whitney test (The error bars represent

the SEM; *n = 9* of different measurements). **E** THP-1 cells, stimulated with LPS, were analyzed over time. Cell and EV lysates, separated on SDS-PAGE, were stained for phosphoproteins *(left panel)* and total protein content *(right panel)*. The data represent a representative experiment from two independent experiments. **F** Lysates from LPS-stimulated THP-1 cells and LPS-EVs were separated on SDS-PAGE and then probed with anti-phospho-tyrosine antibody. The data represent a representative experiment from three independent experiments. **G** Immunoprecipitation of IL1R1 receptor from LPS-stimulated THP-1 cell lysates and LPS-EV lysates was performed. Samples were probed for pIRAK4 and MyD88 antibodies, indicating receptor-associated signaling. The data represent a representative experiment from three independent experiments. **H** Lysates from LPS-stimulated THP-1 cells and LPS-EVs were separated on SDS-PAGE and probed for pIRAK4, MyD88, pTRAF2, and TRADD antibodies, revealing in vitro signaling dynamics. The data represent a representative experiment from two independent experiments.

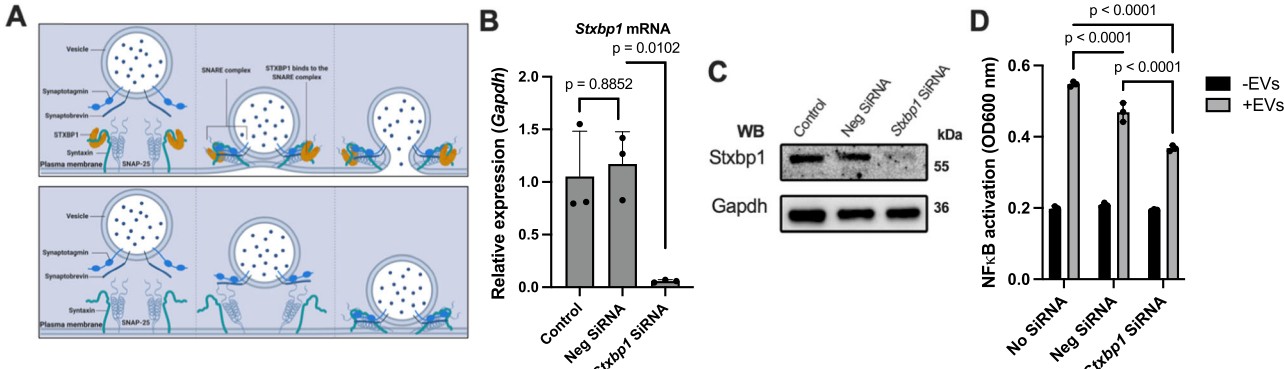

**Fig. 2 | STXBP1 inhibition reduces the ability of EV to fuse with other cells. A** The schematic diagram shows the importance of SNARE complex in cell fusion *(top)*. The EV-cell signaling can be blocked by the inhibiting of Stxbp1 *(bottom)*. **B, C** siRNA targeted against the coding region of the mouse *Stxbp1* gene in RAW-Blue™ cells was used. After 72 h, the decreased expression of *Stxbp1* mRNA and protein was assessed by RT PCR and Western blot analysis The endogenous GAPDH level was measured as internal control. Statistical analysis employed ordinary one-way

ANOVA with a Tukey's multiple comparisons test (The error bars represent the SEM; *n = 3*, of different measurements). **D** *Stxbp1* siRNA-treated RAW-Blue™ cells were stimulated with LPS-EVs *(isolated from 1 × 10⁶ RAW-Blue™)*. NF-κB activation in cell supernatants was measured. Negative siRNA and treatment in the absence of siRNA were used as controls. Statistical analysis employed ordinary one-way ANOVA with a Tukey's multiple comparisons test (The error bars represent the SEM; *n = 3*, of different measurements).

mobilization was also decreased, but not to such an extent as seen with *Stxbp1* mRNA.

To confirm these findings in an in vivo model, mice were administered two doses of liposomal *Stxbp1* siRNA or control siRNA 48 and 24 h prior to LPS-EV treatment as illustrated in Fig. 3A. 4 h after the final treatment, animals were sacrificed and plasma samples were collected for microarray analysis. The results presented as heatmap (Fig. 3B), spider web chart (Fig. 3C), and staple diagrams of selected cytokines (Fig. 3D) show that nearly all inflammatory mediators were upregulated of in LPS-EVs challenged mice that were treated with control siRNA, or when compared with control mice (no treatment with LPS-EVs and siRNA). When mice, however, were treated with *Stxbp1* siRNA a general reduction in the inflammatory response was measured in LPS-EV-injected mice (Fig. 3D). To further investigate the effect of *Stxbp1* siRNA in vivo, lung tissue from the LPS-EVs-treated mice was subjected to RT-PCR and Western blot analysis. Also, here we found that the targeted gene/protein levels were reduced in *Stxbp1* siRNA treated mice compared to the levels seen in mice receiving control siRNA (Supplementary Fig. 1). Our in vitro experiments reveal that both RAW and THP1 cells express STXBP1 under normal conditions, whereas both cell lines downregulate STXBP1 upon stimulation with LPS and LTA as shown by Western blot experiments (Supplementary Fig. 2).

Previous results have shown that the number of EVs in sepsis patients is drastically increased[19]. We therefore employed a murine cecal ligation and puncture (CLP) model to study the role of Stxbp1 under in vivo conditions. While siRNA is a powerful tool for gene

silencing, its use in a CLP model is limited because of the challenges concerning the siRNA delivery, the complexity of the septic response, and problems regarding the timing of the disease progression. We therefore decided to work with genetically modified mice. Since *Stxbp1*⁻/⁻ embryos die immediately after birth[20], we used *Stxbp1*⁺/⁻ mice, for studying the role of Stxbp1 in a CLP model. Abdominal sepsis was induced by puncture of the cecum, *Stxbp1*⁺/⁻ and *Stxbp1*⁺/⁺. After 16 h animals were sacrificed and the cytokine response was measured. As presented in Supplementary Fig. 3, the levels of all cytokines except for IL-6 were reduced in STXBP1⁺/⁻ mice compared to *Stxbp1*⁺/⁺ mice. In summary, our results show that the ability of EVs to initiate inflammatory reactions is reliant on their fusion with target cells.

## The fusion of EVs with cell membranes and the subsequent formation of receptor protein complexes are critical in EV-mediated cell stimulation

Y-27632 is a Rho kinase inhibitor involved in promoting actin cytoskeleton remodeling and cytoplasmic contraction in vitro, which has also been found to facilitate cell fusion events[21]. To investigate whether the inhibitor can also increase cell activation upon EV-cell fusion, RAW-Blue™ cells were treated with Y-27632 (10 μM) for 1 h (Fig. 4A). This was followed by an overnight incubation with EVs and a determination of NF-κB activation. Figure 4B shows that in the absence of EVs (baseline level) no difference in the activation of NF-κB was observed in Y-27632-treated and non-treated cells. When, however, RAW-Blue™ cells were stimulated with EVs, a NF-κB activation was noted in the absence of the

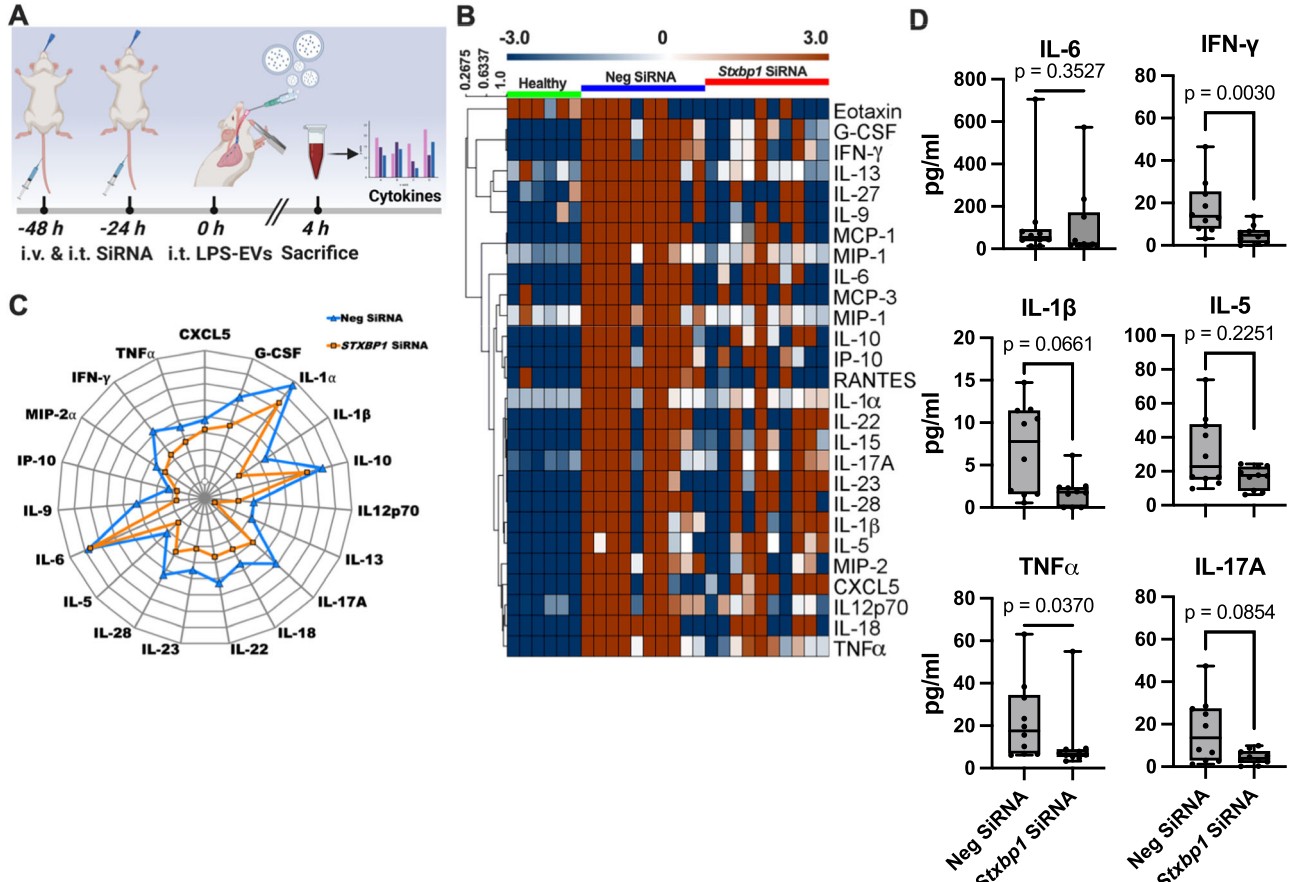

**Fig. 3 | In vivo blocking of Stxbp1 modulates the EV-induced inflammatory response. A** The administration route of siRNA and LPS-EVs followed the subsequent analysis scheme is depicted. **B** LPS-EVs *(from 200 x 10⁶ RAW-Blue™ cells)* were injected per mouse. 4 h after EV-treatment mice were sacrificed, plasma samples were analyzed by multiplex immunoassay, and the results for some selected inflammatory mediators are represented as a heat map analysis. Plasma samples of non-challenge mice or mice treated with neg siRNA served as control. The spider web chart (**C**) and staple diagram (**D**) illustrate the relative up- and down-regulation of some of the selected proteins (*n* = 10 mice per group). The significance was determined using an one-way ANOVA unpaired t-test with a Mann-Whitney test and box-and-whisker plot from minimum value and up to the maximum value. The error bars represent the standard error of the mean.

inhibitor. Activation was even further increased when cells were stimulated with EVs in the presence of Y-27632. To confirm these findings in an in vivo model, mice were intratracheally (*i.t.*) injected with PBS (control) or Y-27632 (10 mg/kg) for 1 h, followed by a four-hour *i.t.* challenge with EVs. Plasma was then collected and subjected to cytokine analysis. As shown in Fig. 4C, the EV treatment induced an increase in IL-6, TNFα, and IL-5 levels, which were higher in Y-27632-treated animals.

The induction of inflammatory mediators such as cytokines is regulated by transcription factors such as NF-κB and AP-1. These factors are activated by adapter proteins, such as MyD88, IRAK4, TRAF2, and TRADD, which interact with the cytoplasmic domains of specific cell surface receptors, thereby allowing for the transduction of extracellular signals into intracellular responses[22]. As our results show that EV fusion with RAW-Blue™ cells leads to a mobilization of NF-κB to the nucleus, we postulated that EVs also unload their adapter proteins upon fusion. To test this hypothesis, we performed a series of NF-κB activation assays as depicted in Fig. 4D. Our results demonstrated that treatment of THP-1 cells with LPS-EVs leads to the mobilization of the transcription factor, which is decreased to background levels when the surface proteins of LPS-EVs were removed by a trypsinization step. THP-1 cells treated with LPS served as control (Fig. 4E). To study the involvement of adapter proteins further we employed THP-1 and THP-1-MyD88 knockout cells. While LPS (a Toll-like receptor 4 agonist) and IL-1β induce NF-κB mobilization via MyD88, TNFα employs TRAF2 for this purpose[23]. This was also confirmed in our results shown in Fig. 4F

(*left panel*). We found that all three stimulants (LPS, TNFα, and IL-1β) can trigger NF-κB mobilization in THP-1 wildtype cells, whereas no activation was measured when LPS and IL-1β were used to stimulate THP-1-MyD88 knockout cells. As expected TNFα-induced NF-κB mobilization was not blocked in THP-1-MyD88 knockout cells. We next tested the NF-κB activating ability of EVs isolated from THP-1 cells that were stimulated with LPS, TNFα, and IL-1β, respectively. Fig. 4F (*right panel*) shows that this treatment resulted in a significant but not total stimulation of the NF-κB pathway regardless of the type of EV used. These findings suggest that EVs, despite the mode of activation of their cell of origin, can mediate MyD88 dependent signaling of the NF-κB pathway in their target cells.

Based on our findings, demonstrating that the fusion of EVs with otherwise non-responsive cells can make them susceptible to a given stimulus, we investigated whether blocking this process could dampen the activation of an inflammatory response. As EV fusion involves a rearrangement of the cytoskeleton, we employed cytochalasin D, an inhibitor of actin polymerization[24]. In addition heparin was used in order to block the docking and binding of EVs to their target cells. Figure 4G shows that both cytochalasin D and heparin prevented LPS-EV triggered activation of the NF-κB pathway in THP-1 cells. To verify our results in an in vivo model, mice were *i.t.* injected with LPS-EVs in the presence or absence of heparin. After 4 h blood samples were recovered and their IL-6 and TNFα concentrations were analyzed. Figure 4H shows that the heparin treatment dampened the inflammatory response otherwise caused by EVs, as seen by decreased levels

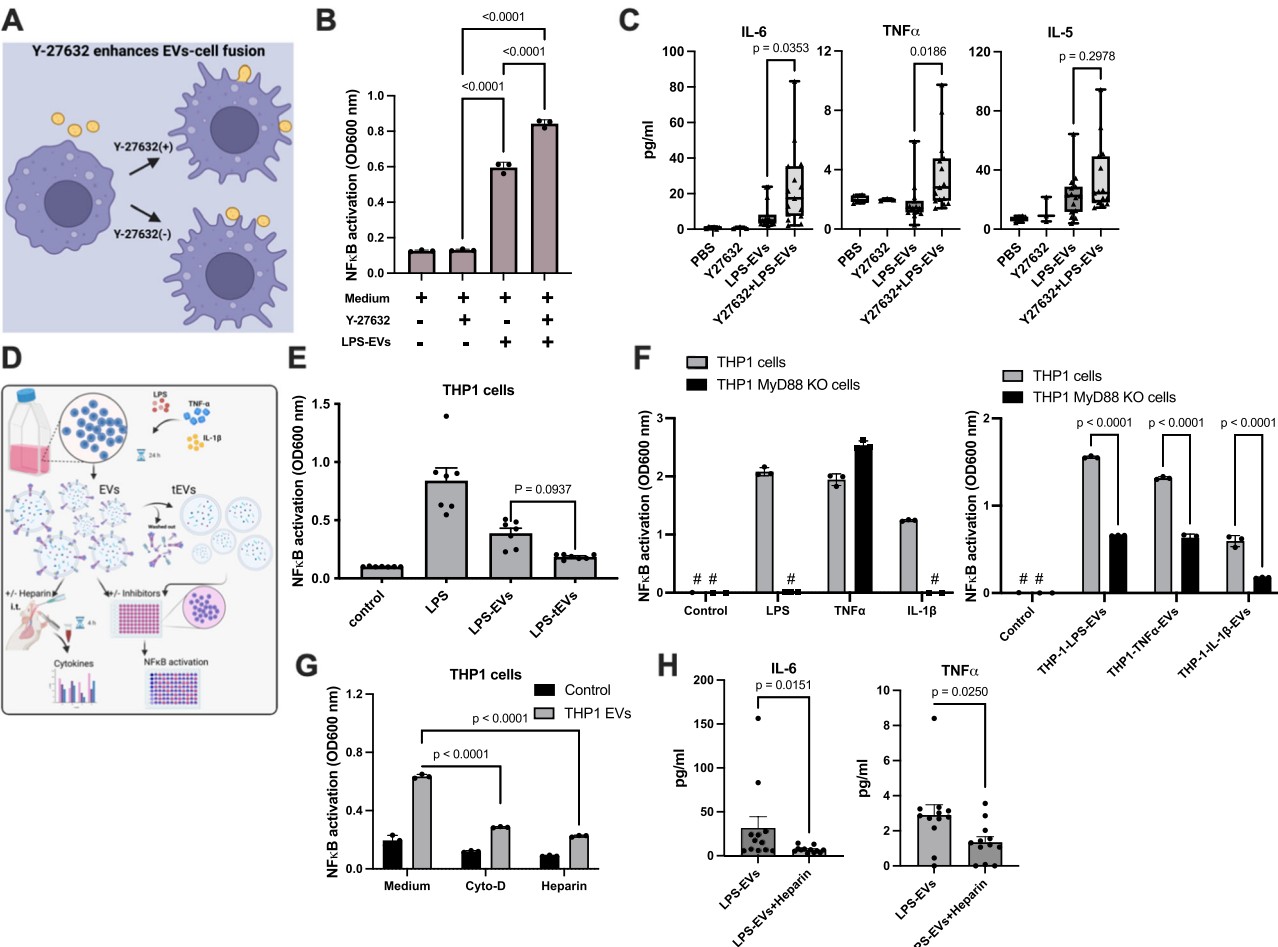

**Fig. 4 | EV fusion and transfer of receptor complexes into recipient cells under in vitro and in vivo conditions. A** Schematic illustration of Y-27632 on EV-cell fusion. **B** Y-27632-treated or non-treated RAW-Blue™ cells were stimulated with LPS-EVs. NF-κB stimulation was measured after overnight incubation. *P* values were determined by ordinary one-way ANOVA with Tukey's multiple comparisons test. Error bars represent SEM. **C** Mice, challenged with LPS-EVs (*from RAW-Blue™ cells*), were treated with or without Y-27632. After 4 h, blood samples were collected and analyzed for IL-6 and TNFα levels. LPS-EVs (*from 200 x 10⁶ RAW-Blue™ cells*) were injected in mice (*LPS-EVs = 15, Y-27632 + LPS-EVs = 15, mice per group*). PBS- and Y-27632-treated mice without EVs served as controls (*PBS n = 6 and Y-27632 n = 3*). Statistical analysis utilized ordinary one-way ANOVA with Tukey's multiple comparisons test, results presented as box-and-whisker plots from minimum value and up to the maximum value (SEM). **D** Schematic diagram highlighting the importance of EV outer bound proteins in host cell stimulation. EV-cell signaling can be blocked by eliminating surface receptors or using inhibitors (Created with BioRender.com). **E** THP-1 cells treated with LPS, LPS-EVs, and trypsinized surface proteins of LPS-EVs (*LPS-tEVs*) were stimulated, and NF-κB stimulation was measured. Statistical analysis employed ordinary one-way ANOVA with a Tukey's multiple comparisons test (SEM; *n = 3* of different measurements). **F** THP-1 cells and MyD88 knockout THP-1 cells were treated with LPS, TNFα, and IL-1β. NF-κB stimulation was measured (*left panel*). EVs from overnight stimulation with LPS (*LPS-EVs*), TNFα (*TNFα-EVs*), and IL-1β (*IL-1β-EVs*) were used to stimulate cells (*right panel*). Statistical analysis employed ordinary two-way ANOVA with a Tukey's multiple comparisons test (The error bars represent the SEM; *n = 3* of different measurements). **G** Ability of LPS-EV stimulation (1 × 10⁶ cells per reaction) in the presence of cytochalasin D and heparin was tested. Statistical analysis employed ordinary two-way ANOVA with a Tukey's multiple comparisons test (SEM; *n = 3* of different measurements). **H** Mice challenged with LPS-EVs with or without heparin (100 μg/ml). After 4 h, blood samples were collected and analyzed for IL-6 and TNFα levels. Statistical analysis employed ordinary one-way ANOVA with a Tukey's multiple comparisons test (SEM; *n = 12* mice per group).

of IL-6 and TNFα in plasma samples of the challenged mice. Based on these findings we concluded that EVs trigger inflammatory reactions which can be modulated by preventing their interactions with the target cells.

## EV-mediated cell stimulation in transfected HEK-Blue™ reporter cells

We next investigated whether EVs can prime cells that are not responsive to a given stimulant. To test this we switched to HEK293 cells, which is an immortalized human embryonic kidney cell line. To confirm that EVs bind to HEK293 cells under our experimental conditions we employed SEM (scanning electron microscopy). Supplementary Fig. 4 depicts HEK293 cells in the absence of LPS-EVs (*left panel*). When incubated with LPS-EVs, membrane ruffling of HEK293 cells was

noted around the EV attachment site, suggesting the fusion has been initiated (*middle panel*). This process resembles merging of LPS-EVs with their target cells according to the Ω-profile as described by Wen et al.[25]. Having demonstrated that EVs are able to fuse with HEK293 cells, we employed genetically engineered HEK-Blue™ reporter cells that were transfected with IL1R1 (interleukin 1 receptor, type I), TNFR1 (tumor necrosis factor receptor 1), and IL10-R (interleukin-10 receptor), respectively. Non-transfected HEK-Blue™ reporter cells served as control. In order to prevent potential cross-activation by other ligands, all three cell lines were genetically modified *e.g.* IL-1β HEK-Blue™ reporter cells (IL1R1 transfected HEK293 cells) did not express *TLR3, TLR5* and *TNFR1*, TNFα HEK-Blue™ reporter cells (TNFR1-transfected HEK293 cells) lack the MyD88 gene, and IL-10 HEK-Blue™ reporter cells (interleukin-10 receptor-transfected HEK293 cells) did not express

*TLR2* and *TNFR1*. A schematic outline of the experiments with the three cell lines is presented in Fig. 5A. Using this approach we found that all transfected cell lines only responded to their corresponding agonist, while no NF-κB activation was recorded when stimulated with other ligands. In addition it was found that the three transfected cell lines as well as the non-transfected HEK293 cells were not responsive to stimulation with LPS.

HEK-Blue™ reporter cells were also stimulated with their specific agonist (TNFα, IL-1β, and IL-10). After overnight incubation EVs (in the following referred to as TNFα-EVs, IL-1β-EVs, and IL-10-EVs) were isolated and used to stimulate the three reporter HEK-Blue™ cell lines. EVs were also treated with trypsin, to remove all surface proteins, followed by an additional washing step before incubating with the three reporter cell lines. Fig. 5C (*upper panel*) shows that TNFα-EVs, evoked an NF-κB activation in TNFα reporter cells. Treatment with trypsin led to a reduction of signal intensity. In addition, also IL-1β-EVs and IL-10-EVs induced an NF-κB mobilization in TNFα reporter cells, which was down-regulated when EVs were subjected to trypsin treatment. Similar results were obtained when IL1R1- (middle panel) and IL10-tranfected reporter cells (lower panel) were incubated with TNFα-EVs, IL-1β-EVs, and IL-10-EVs.

All together the results show that EVs are able to render HEK293 cells susceptible to an otherwise non-stimulating ligand. Moreover, our findings further suggest that upon fusion EVs can translocate their surface receptors to the cell membrane of target cells. To confirm this conclusion, IL-1β HEK-Blue™ reporter cells were incubated with TNFα-EVs. To this end, we determined the levels of TNFα and IL-1β in TNFα-EVs and IL-1β-EVs, respectively. Fig. 5D shows that TNFR1 is found in TNFα-EVs and IL-1β-EVs, while IL-1β-EVs was only detected in IL-1β-EVs. Next we measured if upon fusion of IL-1β-EVs and TNFα-EVs the respective receptors are enriched in the cytosol or plasma membrane of the targeted cell. Fig. 5D (left panel) shows that upon fusion, an increase of TNFR1 in the cell membrane but not the cytosol was recorded. Similar results were obtained when TNFα HEK-Blue™ reporter cells were treated with IL-1β-EVs (Fig. 5D right panel). In conclusion, our results show that the fusion EVs results in a translocation of surface bound receptor proteins into the cell membrane and a transfer of intracellular adaptor proteins into the cytosol of HEK293.

## Agonist-binding to the receptors on EVs is important for cell stimulation

In our experiments EVs were generated by stimulation of the three HEK-Blue™ reporter cell lines with their respective agonists (TNFα, IL-1β, and IL-10). These experimental conditions indicate that the agonists remain attached to their respective receptors at the surface of the target cells upon fusion with EVs. Therefore, no additional agonist stimulation should be required to trigger intracellular signaling in these cells. To test this, we used neutralizing antibodies against TNF-α and IL-1β and performed experiments as outlined in Fig. 6A. Fig. 6B (left panel) shows that stimulation with TNFα led to an NF-κB-activation in TNFα HEK-Blue™ reporter cells which was down-regulated when a neutralizing anti TNFα-antibody was administered. No inhibition of an NF-κB-mobilization was seen when the neutralizing anti TNFα-antibody was replaced with an IgA2 isotype control antibody. TNFα HEK-Blue™ reporter cells, treated with IL-1β in the presence or absence of an IL-1β neutralizing antibody or an IgG isotype control antibody, were used as negative control. As expected, NF-κB-activation was not recorded under these experimental conditions. Experiments were also performed using IL-1β HEK-Blue™ reporter cells. Cells were incubated with TNFα and IL-1β, as well as with their respective neutralizing antibodies and isotype controls. Fig. 6B (middle panel) shows that NF-κB-activation was not observed when cells were incubated with TNFα in the presence or absence of a TNFα neutralizing antibody or an IgG isotype control. However, when cells

were treated with IL-1β, an NF-κB-mobilization was seen that was partially blocked by an IL-1β neutralizing antibody, but not by an IgG isotype control antibody. Finally, IL-10 transfected HEK-Blue™ reporter cells were subjected to the same experimental protocol, which did not result in an activation of NF-κB upon stimulation with TNFα and IL-1β in the presence or absence of respective neutralizing antibodies and isotype controls (right panel).

The activation of NF-κB was also tested using EVs isolated from stimulated HEK-Blue™ reporter cells. The results in Fig. 6C (left panel) show TNFα-EVs were also capable of triggering NF-κB-activation in TNFα HEK-Blue™ reporter cells. Unlike the experiments performed with TNFα (Fig. 6B; left panel) neutralizing anti TNFα antibodies were unable to block NF-κB-activation. In addition, an NF-κB mobilization was observed when in TNFα HEK-Blue™ reporter cells were incubated with IL-1β EVs. Also here activation was not blocked by IL-1β neutralizing antibodies. Similar results were obtained the experiments were performed with IL-1β HEK-Blue™ reporter cells (middle panel) and IL-10 HEK-Blue™ reporter cells (right panel).

To ensure that the signals observed were not due to a contamination in the EV-preparation, EV-free supernatants were employed. Fig. 6D (left panel) shows that the medium alone (control) was not able to activate an NF-κB mobilization in TNFα HEK-Blue™ reporter cells. When supernatants from a TNFα HEK-Blue™ reporter cell overnight culture, stimulated with TNFα, were added to a new TNFα HEK-Blue™ reporter cell culture, NF-κB activation was measured, which was blocked by the neutralizing anti TNFα-antibody. It should be noted that supernatants from IL-1β and IL10 HEK-Blue™ reporter cells that were triggered with IL-1β and IL-10, in the presence or absence of their respective neutralizing antibody, induced only a low-level of NF-κB activation under these experimental conditions. When the experiments were performed in IL-1β HEK-Blue™ reporter cells (middle panel) or IL-10 HEK-Blue™ reporter cells (right panel), NF-κB activation was only seen when cells were stimulated with their respective agonists (IL-1β and IL-10). In both cases NF-κB mobilization was inhibited by the respective neutralizing antibodies. Based on these findings it can be concluded that cells which fuse with EVs, already carrying a receptor in complex with its agonist, do not require additional agonist stimulation for initiating intracellular signaling cascades.

The next series of experiments was conducted to study the kinetics of NF-κB activation by TNFα-EVs and IL-1β-EVs in a time-dependent manner. As shown in Fig. 6E incubation of TNFα-EVs (left panel) and IL-1β-EVs (right panel) induced an NF-κB activation in their respective responder cells after only 4 h of incubation, which was not observed when cells were stimulated with TNFα or IL-1β alone. In cells treated with TNFα-EVs, no further NF-κB activation was observed after 16 h of incubation (Fig. 6E, *left panel*), whereas cells treated IL-1β-EVs continued to have an increased NF-κB response (Fig. 6E, right panel). These results provide evidence that EVs can activate an inflammatory response by fusing with non-responsive cells. Because EV-fusion leads to a translocation of agonist-bound receptors to the cell membrane and a transfer of signaling adapter proteins into the cytosol of the targeted cells, the signal cannot only be blocked by neutralizing antibodies, but it also triggers a faster inflammatory response as seen when stimulated with the agonists only.

## Injection of EV triggers an inflammatory response in mice

For the next series of experiments mice were *i.t.* injected with TNFα-EVs. Thirty min after EV injection, animals were sacrificed, lungs were recovered, and subjected to immunofluorescence microscopy. Supplementary Fig. 5 (upper panel) depicts a lung section from a control mouse. When lung biopsies from TNFα-EVs -treated mice were analyzed, EVs were found attached to cells in the airways (lower panels). The interaction of TNFα-EVs with immune cells under in vivo conditions was also investigated by confocal immune microscopy (Supplementary Fig. 6), where we found that injected TNFα-EVs are associated

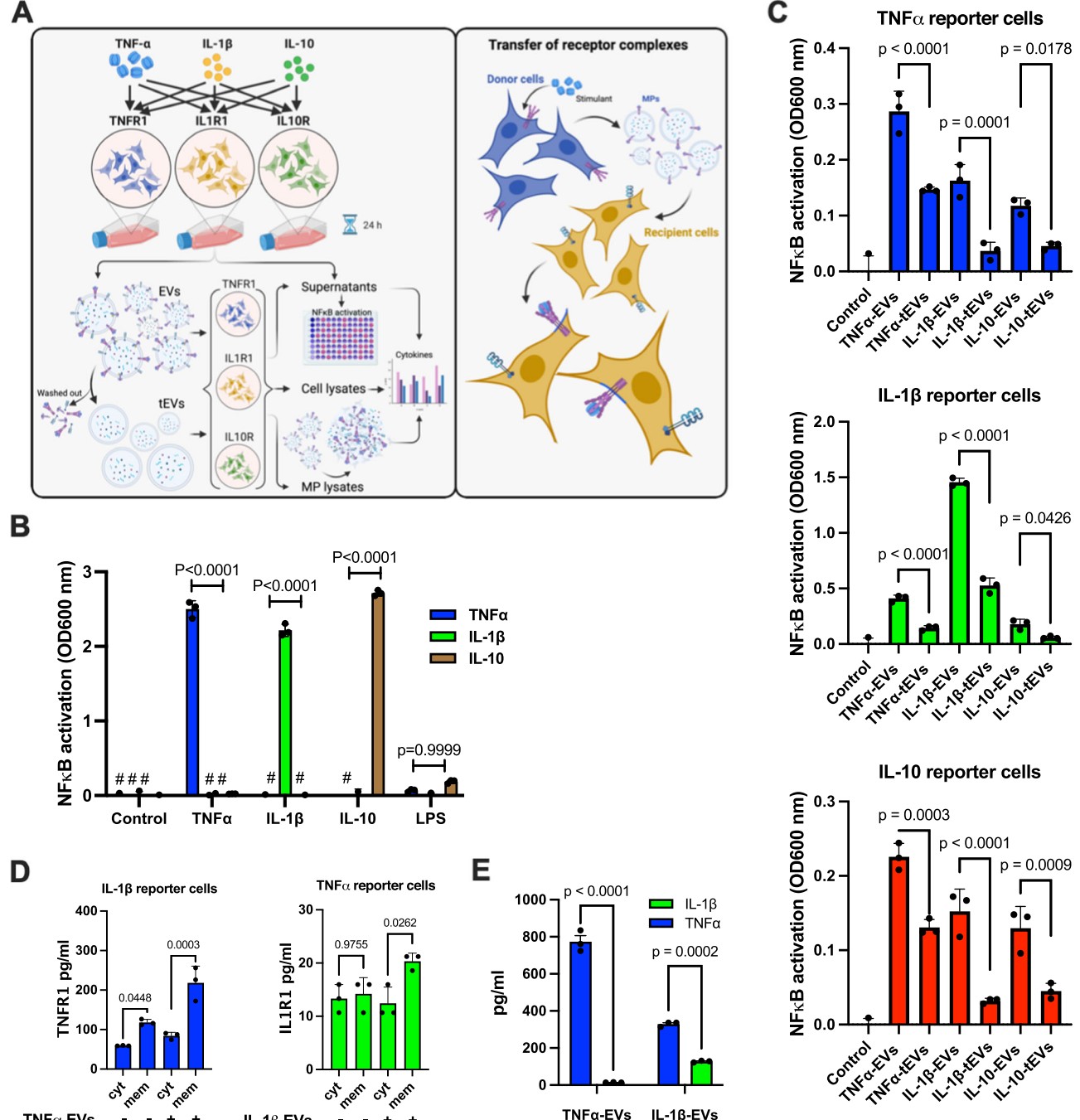

**Fig. 5 | Mechanism of EV-mediated cell stimulation.** To examine the role of EV-mediated cell stimulation through receptor protein signaling, HEK-Blue™ cells were transfected with 1) TNFR1, the receptor for TNFα, 2) IL1R1, the receptor for IL-1β, and 3) IL10-R, the receptor for IL-10. **A** The schematic diagram shows the transfer of ligand-receptor immune complexes to the recipient cells (Created with BioRender.com). **B** The three reporter HEK-Blue™ cell lines were stimulated with TNFα, IL-1β, IL-10, and LPS *(negative control)* and after an overnight incubation NF-κB stimulation was measured. *P* values were determined using an ordinary two-way ANOVA with a Tukey's multiple comparisons test using GraphPad Prism software. The error bars represent SEM. **C** EVs were generated by stimulation of HEK293 reporter cells with TNFα, IL-1β, and IL-10. The collected EVs were then incubated with the TNFα, IL-1β, and IL-10 HEK-Blue™ reporter cell lines. Alternatively, EVs after a trypsinization and washing step were used to stimulate the three reporter cell lines. EVs from a total of $1 \times 10^6$ stimulated HEK-Blue™ reporter cells were used per reaction. *P* values were determined using an ordinary one-way ANOVA with a Tukey's multiple comparisons test using GraphPad Prism software. The error bars represent SEM. **D** IL-1β-HEK-Blue™ cells were incubated with TNFα-EV *(from 1 x 10⁷ stimulated TNFα-HEK-Blue™ cells)* for 30 min and transfer of the TNFR1 to the cell membrane and cytosol was determined *(left panel)*. TNFα-HEK-Blue™ cells were incubated with IL-1β-EV *(from 1 x 10⁷ stimulated IL-1β-HEK-Blue™ cells)* for 30 min and transfer of the IL1R1 to the cell membrane and cytosol was determined *(right panel)*. The significance was determined using a one-way ANOVA with a Tukey's multiple comparisons test. The error bars represent the standard error of the mean. **E** TNFα and IL-1β receptor levels were measured in EVs collected from the $1 \times 10^7$ TNFR1- and IL-1β-reporter HEK-Blue™ cells. P-values were determined using an ordinary one-way ANOVA with a Tukey's multiple comparisons test using GraphPad Prism software. The error bars represent SEM. In vitro data represent a representative experiment (from three independently performed experiments).

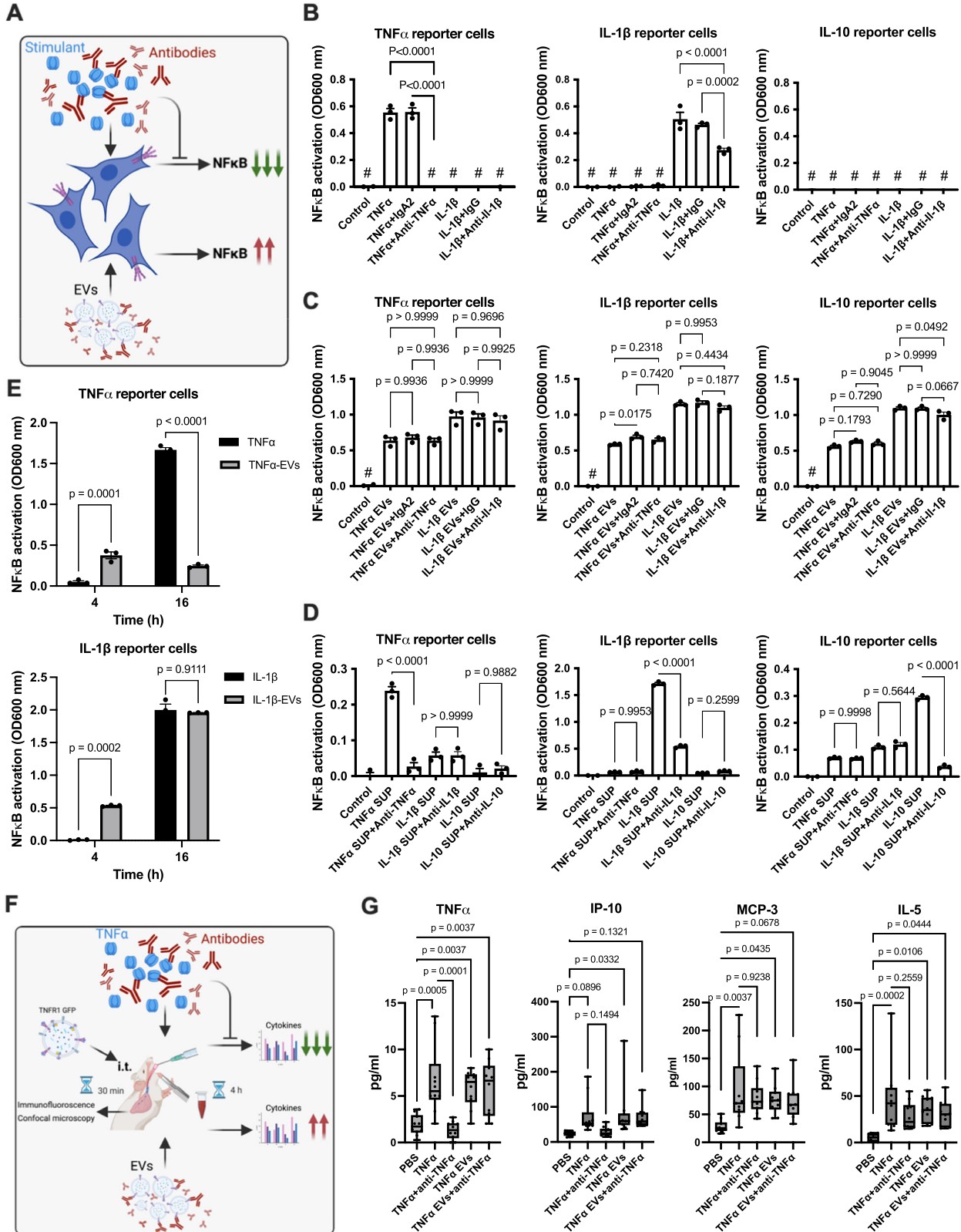

with immune cells. To further investigate the effect of EVs in vivo, mice were challenged (*i.t.*) with TNFα and TNFα-EVs in the absence or presence of the TNFα neutralizing antibody as illustrated in Fig. 6F. 4 h after treatment mice were sacrificed, blood samples were recovered and analyzed for their TNFα, IP10, MCP-3, and IL-5 content. Fig. 6G shows that the TNFα challenge triggered an induction of all six cytokines measured. Their levels were partially down-regulated when the

animals were also treated with the TNFα neutralizing antibody. The treatment with TNFα-EVs also evoked an inflammatory response which was significant, but not as pronounced, as seen when TNFα was administrated. This response was not blocked when the TNFα neutralizing antibody was applied. Together our findings show that the release of EVs under ex vivo and in vivo conditions can contribute to the induction of systemic inflammatory reactions. Moreover our

**Fig. 6 | In vitro and in vivo role of receptor bound ligands for EV-mediated cell stimulation. A** Schematic diagram illustrating the significance of ligand-receptor immune complexes in triggering signaling in recipient cells (Created with BioRender.com). TNFα, IL-1β, and IL-10 HEK-Blue™ reporter cell lines were stimulated with TNFα, IL-1β, and IL-10 (**B**) or LPS-EVs (*from 1 × 10^6 stimulated HEK293 cells*) (**C**) in the presence or absence of neutralizing antibodies. IgA2 and IgG isotype antibodies were used as controls. NF-κB activation was measured after an overnight incubation. P-values determined by ordinary one-way ANOVA with Tukey's multiple comparisons test (SEM; n = 3, of different measurements). **D** TNFα, IL-1β, and IL-10 HEK-Blue™ reporter cell lines were incubated with EV-free supernatants in the presence or absence of neutralizing antibodies. IgA2 and IgG isotype antibodies were used as controls. NF-κB stimulation was determined after an overnight incubation. P-values determined by ordinary one-way ANOVA with Tukey's multiple comparisons test (SEM; $n = 3$, of different measurements). **E** TNFα-HEK-Blue™ reporter cells were incubated with TNFα or TNFα-EVs (*from the 1 x 10^6 TNFα-HEK-Blue™ reporter cells*). IL-1β -HEK-Blue™ reporter cells were incubated with IL-1β or IL-1β -EVs (*from 1 x 10^6 IL-1β-HEK-Blue™ reporter cells*). NF-κB stimulation was measured 4 h and 16 h after stimulation. P values determined by ordinary two-way ANOVA with Tukey's multiple comparisons test (SEM; $n = 3$, of different measurements). **F** The schematic diagram shows the experimental design of the in vivo experiments (Created with BioRender.com). **G.** Mice were challenged with TNFα or TNFα-EVs (*from 200 million THP1 cells/mouse*) in the presence or absence of neutralizing anti-TNFα antibodies (*10 μg/mouse*). An IgA2 antibody (*10 μg/mouse*) served as a negative control. 2 h after treatment, blood samples were collected and analyzed for TNFα, IP10, MCP-3, and IL-5 levels. P-values determined by two-way ANOVA with Tukey's multiple comparisons test. (SEM; $n = 10$ mice per group).

findings further reveal that receptor antagonists such as neutralizing antibodies against inflammatory mediators, can fail to block a receptor-mediated cell activation that it caused by the fusion of EVs.

### Identification and characterization of EVs isolated from polytrauma and sepsis patients

As our in vitro and in vivo results show that EVs released from activated inflammatory immune cells can render non-responsive cell to pro-inflammatory stimuli, our findings suggest that EVs play an important role in severe inflammatory diseases. Notably, polytrauma and sepsis belong to the most severe forms of inflammatory diseases. We therefore decided to use plasma samples from these two patients group for the last set of experiments. When employing a Pro-Q Diamond dye-based SDS-PAGE approach, we found similar phosphorylated protein patterns in EVs isolated from polytrauma and sepsis patients (Fig. 7A). For further characterization of the protein content of EVs from these patients, multiplex immunoassay and quantitative MALDI-Tandem Mass Tag (TMT) analysis was performed (raw data, including method files are available from ProteomeXchange (PXD048039)). Using these techniques 4091 proteins were identified. Further analysis revealed that the concentration of 71 proteins was significantly changed (FDR = 0). In most cases (69 proteins) an up-regulation was noted, whereas only two proteins were found down-regulated (Supplementary Fig. 7A). A summary of the findings is also presented in the Heatmap shown in Supplementary Fig. 7B. Using the *KEGG pathway analysis* program, we found that most of the identified proteins have inflammatory activities and are involved in NF-κB signaling (Fig. 7B).

Based on these results all identified proteins from the MALDI analysis were imported into Enrichr database for functional annotation (GO & KEGG) analysis. Figure 7C depicts the results of the involvement of the 20 most abundant up-regulated proteins in biological, molecular, and cellular pathways, respectively. The complete list of GO terms and pathways is provided in Source data file. Functional biological analysis for these modules indicates that most of the identified proteins play a role in cell-mediated immunity, including immune activation, signaling, protein transport, and cellular localization. Moreover, significantly enriched GO molecular function terms for EV proteins with important features such as mediating cell-cell adhesion, promoting gene regulation, triggering protein dimerization, and evoking protein phosphorylation (Fig. 7C). In addition, cellular functional analysis revealed that many of these proteins are localized in the cytosol, in the cell membrane, and extracellularly (Fig. 7C). Together we show that EVs are equipped with all tools to cause systemic inflammation by fusion with otherwise non-immune responsive cells. As upon fusion receptors complexes carrying their agonists as well as intracellular adaptor proteins are translocated, targeted cells can immediately trigger an induction of an inflammatory response by mobilization of the NF-κB pathways. Additional analysis of EVs of proteins from polytrauma and sepsis patients revealed that most proteins identified play an important role in the induction of inflammatory reactions.

## Discussion

Emerging research is suggesting that EVs have a crucial function in inducing inflammation upon fusion with target cells[26]. EVs have been, for example, found to transfer a range of pro-inflammatory factors, such as cytokines, chemokines, and damage-associated molecular patterns (DAMPs), to recipient cells[27]. Though it is generally accepted that this can trigger immune responses and may contribute to the progression of inflammatory-driven diseases such as polytrauma and sepsis, the molecular mechanisms behind these processes are still not fully understood and require further investigation.

A better knowledge of how cells communicate via the release of EVs is therefore not only critical for gaining insights into fundamental biological processes, but also for the development of novel diagnostic and therapeutic approaches that can target specific diseases and improve patient outcomes. The fusion of EVs with their recipient cells is in particular important as it can induce immune responses during inflammation in otherwise non-responsive cells. Several studies have published an involvement of EVs in various biological processes. For instance, Hung et al. reported in 2019 that tumor-secreted EVs protect EGFR and promote proliferation and migration ability in recipient cancer cells, which suggest a novel mechanism of drug resistance in breast cancer[28]. Jiang et al. published in 2022 that circulating EVs can reduce the size of infarcted tissue and promote the proliferation and function of endothelial progenitor cells via regulating miR-190a-3p/CXCR4/CXCL12 pathway[29] and Zeng et al. described in 2018 that EVs are involved in the transfer of pre-formed antigenic peptides and MHC molecules between cells which has implications in immune homeostasis, tolerance, and regulation of T cell immunity[30].

In the present study, the focus was on the role of EVs in promoting systemic inflammatory responses. Our findings show that upon fusion, EVs can equip their target cells with the molecular machinery that makes them susceptible to an inflammatory signal. This not only includes the translocation of inflammatory receptors, but also the translocation of intracellular signal tranduction adapter protein, such as MyD88, IRAK4, TRAF2, and TRADD, which has not been shown before. Moreover, we found that EVs can transfer receptors that are already in complex with their agonists implying that no further stimulation with an additional exogenous ligand is needed to initiate intracellular signaling events (Fig. 8). Since receptor/agonist complexes can be formed before fusion, the application of receptor antagonists will therefore have only limited effect in abolishing EV-induced inflammatory responses. These findings may explain why clinical trials using anti-inflammatory drugs have failed. Considering that EVs are released into the circulation, the mechanism presented in this study also provides a plausible explanation as to why cells at a non-inflamed side can become responsive in the absence of an inflammatory signal. Together, our results show a fundamental role of EVs in induction of massive inflammatory complications under clinical conditions.

Our experiments using *Stxbp1* siRNA show that the fusion of EVs with their target cells is essential for evoking inflammatory reactions. Thus, the application of heparin and its derivatives in clinical practice

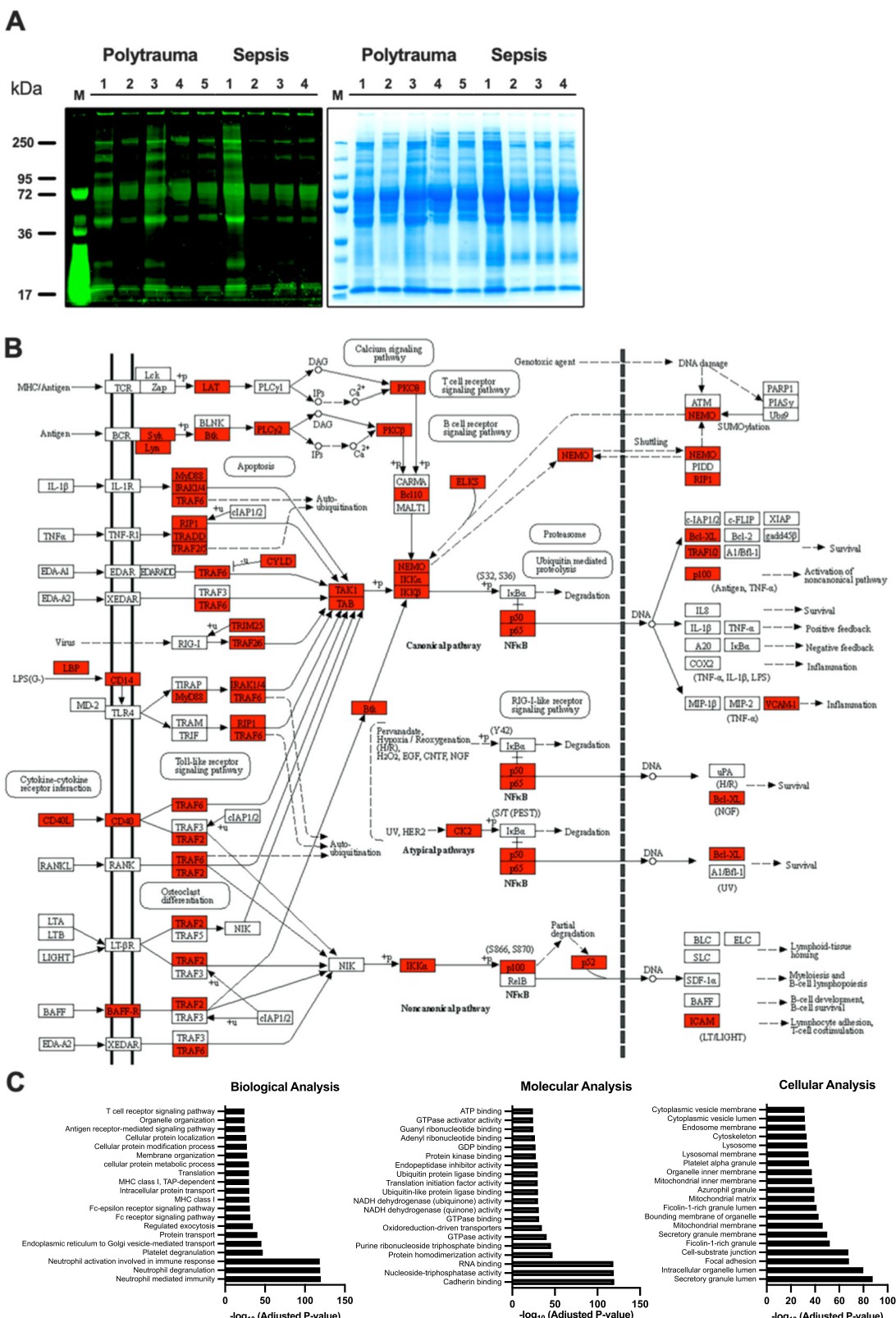

**Fig. 7 | Proteomic profiling and identification of differential expressed proteins in EVs isolated from polytrauma and sepsis patients. A** EV lysates were separated on SDS-PAGE, stained for phosphorylated proteins using Pro-Q Diamond staining *(left panel)* and total proteins were stained using Gel Code Blue Safe Protein Stain *(right panel)*. Polytrauma (*n = 5*) and sepsis patients (*n = 4*). **B** NF-κB pathway involved EV identified proteins in MALDI were highlighted in red, using KEGG pathway analysis *(pooled from 5 individual samples)*. **C** Gene ontology enrichment analysis of total identified proteins in MALDI-TMT analysis is shown. GO biological process, molecular process, and cellular component, enrichment analyses were performed using Enirchr web-based software, with the top 20 enriched GO terms displayed. Pooled EVs isolated from healthy individuals were used for normalization. The adjusted *p* value is computed using the Benjamini-Hochberg method for correction for multiple hypotheses testing.

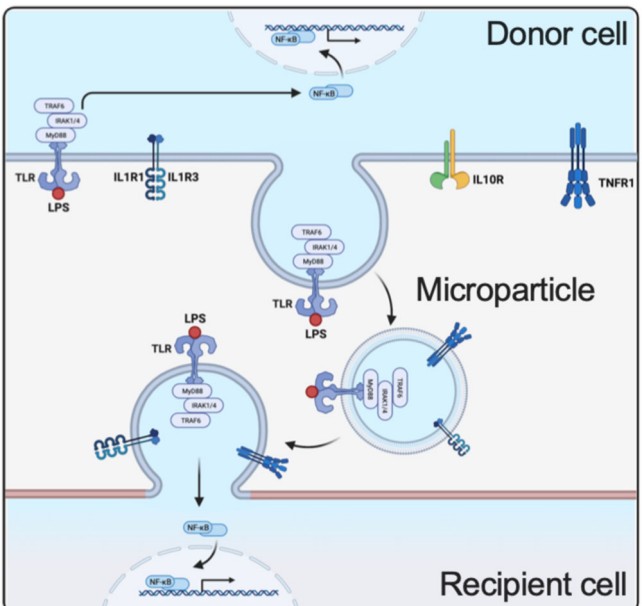

**Fig. 8 | The schematic diagram shows the transfer of ligand-receptor immune complexes to the recipient cells and its importance in EV signaling.** Binding of LPS to a donor cell via TLR initiates intracellular signaling pathways, engaging MYD88, IRAK1/4, TRAF6, and NF-kB. Subsequently, microvesicles containing the complete signaling complex are released from the donor cell. When these microparticles bind to their target, the recipient cell can leverage the signaling complex to activate its NF-kB pathway. (Created with BioRender.com).

offers a promising option for developing novel anti-inflammatory and anticoagulant therapies, which in addition can prevent the binding and subsequent fusion of EVs with the targeted cell. Notably, heparin is already routinely used to treat conditions such as acute venous thromboembolism[31,32] and pulmonary embolism[33], as well as in patients undergoing cardiopulmonary bypass operation[34]. Other studies suggest that the anti-inflammatory activity of heparin can also be used to treat asthma patients, patients undergoing cardiopulmonary bypass, and patients suffering from inflammatory bowel disease, although the beneficial effects still need to be confirmed in larger clinical studies[35]. In addition to heparin, also cytoskeleton remodeling inhibitors, such as cytochalasin D, are interesting targets for drug development. However, as these substances may evoke unwanted side effects, they are therefore less attractive candidates. This likely also applies to other inhibitors that can modulate cytoskeleton rearrangement. Therefore, blocking fusion between EVs and their target cells, rather than inhibition of cytoskeleton rearrangements, appears to be an interesting approach to prevent or inhibit the induction of systemic inflammatory complications.

The identification of novel modes of actions that can prevent the fusion of EVs with other cells is a promising concept for the development of novel anti-inflammatory therapies. This applies not only to the treatment of the described clinical conditions, *i.e.* polytrauma and sepsis, but also to other diseases. For instance, application of heparin has been reported to possess clinical benefit in the treatment of patients with cancer[36,37], where the generation of EVs has been described contribute to the pathogenesis of cancer[38]. This mechanism has also been described to prevent the formation of metastases[11]. Elevated concentrations of EVs have also been reported in patients with SARS-CoV-2 infections[39]. In this study it was found that EV levels are associated with the severity of the disease and correlate with mortality. Importantly, treatment with heparin has been described to reduce mortality in a non-randomized clinical study involving patients with COVID-19 by counteracting the procoagulant initial state of the infection[40,41]. In another recent report Xia et al. show SARS-CoV-2 is

using extracellular vesicles to infect other cells[42]. The authors also found that this process that cannot be blocked with SARS-CoV-2 neutralizing antibodies[42]. These findings are in line with our results as they show that the cargo that is translocated by EVs is protected despite applying soluble factors, such as antibodies, receptor antagonists, or other small molecules that would otherwise act as antagonists. Moreover they confirm our conclusion that approaches which block the fusion of EVs with recipient cells for instance with heparin, are a promising strategy to prevent the induction of systemic inflammatory reactions.

Polytrauma and severe infectious diseases including sepsis remain a major clinical problem even though major advances in intensive medical care have been made in recent years. Both clinical conditions are evoked by a dysregulated induction of pro- and anti-inflammatory cascades often triggered by systemic reactions to tissue trauma or microbial stimuli. A number of clinical trials with anti-inflammatory drugs, such as a monoclonal antibody against TNFα and an interleukin-1 receptor antagonist have been conducted to counteract the pathological inflammatory response in bacterial sepsis with little to no success[43,44].

To conclude, targeting the fusion between EVs with their recipient cells provides a novel approach for treating these life-threatening conditions. By inhibiting the fusion of EVs with recipient cells with heparin, the transfer of pro-inflammatory molecules can be blocked and the spread of inflammation throughout the body can be limited. In clinical studies, heparin has been shown to reduce mortality in patients with sepsis and COVID-19 infections, suggesting that targeting EV fusion may be a promising approach for the development of new treatments for these conditions.

## Methods
### Ethical statement
Informed consent was obtained from the patients, or the guardians included in this study. Samples from Helsingborg Hospital ICU were approved by the Ethics committee in Lund Dnr. 2015/467 and 2019/04558 and samples from Charles-University Prague were approved by the Ethics Committee of the Military University Hospital Prague Dnr. 108/9-36/2016-UVN. All experiments on mice were performed according to Swedish Animal Welfare Act SFS 1988:534 and were approved by the Animal Ethics Committee of Malmö/Lund, Sweden (permit numbers Dnr 5.8.18-18-01753/2022). Mice were housed in Innovive IVC Rodent Caging System on a 12/12 h light/dark cycle. The ambient temperature ranged between 19 and 23 °C with humidity 55 ± 10%. All animals had free access to water and chow. Technicians and staff did not enter the room during the dark cycle unless strictly required to collect cages for health monitoring. All animals received care according to the USA Principles of Laboratory Animal Care of the National Society for Medical Research, Guide for the Care and Use of Laboratory Animals, National Academies Press (1996).

**Materials.** Reagents, proteins, and antibodies: Recombinant human TNFα (rcyc-htnfa), IL-1β (rcyec-hil1b), anti-human TNFα (htnfa-mab7), anti-human IL-1β (mabg-hil1b), IgA2 (bgal-mab7) and IgG (mabg1-ctrlm) were purchased from InvivoGen. Recombinant human IL-10 (217-IL-010) and anti-human IL-10 (AF-217) antibodies were obtained from R&D Systems. Y-27632 (HY-10071), Cytochalasin D (Hy-17567B) and heparin (Hy-N6682) were from MedChemexpress.

### Flow cytometry
CD42b receptor transfer to neutrophils and monocytes cell surface was investigated by flow cytometry using a Beckman Coulter Cytoflex (Beckman). Citrated blood from five healthy donors was incubated with 0.5 μg/ml *E. coli* O111:B4 LPS (Sigma-Aldrich®) for 15 min at 37 °C. 100 μl of blood was used and red blood cells were lysed with a BD Phosflow™ Lyse/Fix 5X. After lysis, samples were washed once with

0.5% BSA in PBS and incubated with CD66b-FITC (Clone: G10F5, BD Bioscience), CD42b-PE-Cy™5 (Clone: HIP1, BD Bioscience) and CD14-BV421 (Clone: HCD14, BioLegend®) antibodies were added (1:50) to the cell suspension and incubated for 60 min at 37 °C in the dark. CD42b is platelet-specific surface protein as claimed by the manufacturer. To exclude that other cells can up-regulate CD42b upon stimulation, isolated neutrophils were stimulated with LPS. For one donor, the manual gating analysis using FMO, and gating controls was carried out. Samples were washed once with 0.5% BSA in PBS and the cell pellet was resuspended in 300 µl of washing buffer. The percentage of CD42b on the cell surface was calibrated with control cells that were not treated with LPS. The results are presented as mean values ± s.e.m.

### Stxbp1 RNA expression analysis
To analyze *Stxbp1* gene expression in the lungs of healthy, siRNA injected mice, RAW-Blue cells, and THP-1 cells, RNA was isolated followed by DNase digestion, cDNA synthesis and quantitative PCR analysis (Supplementary Fig. 1).

### RNA isolation and DNase digestion
For analyzing relative *Stxbp1* RNA expression by quantitative polymerase chain reaction (qPCR), livers were transferred from *RNA*later™ into a 2.0 ml SC Micro Tube PCR-PT containing 600 µl of RLT Plus buffer including 1 % of β-mercaptoethanol and 10 to 15 1.4 mm ceramic beads (QIAGEN). Lung/Raw-Blue cells were homogenized in this solution for 30 s at 7000 rpm using a MagNA Lyser Instrument Version 4.0 (ROCHE). RNA was isolated following the protocol of RNeasy® Plus Mini Kit (QIAGEN) and eluted with 50 µl of nuclease-free water (Ambion®). Furthermore, a NanoDrop® Spectrophotometer ND-1000 was used to determine RNA concentration and purity. To exclude DNA contamination, further DNase digestion by using Deoxyribonuclease I, Amplification Grade (18068015, Invitrogen by Thermo Fisher Scientific) according to manufacturer's protocol was performed. Reaction mixture served as template for further complementary DNA (cDNA) synthesis.

### Complementary DNA Synthesis
One µl of recent DNase digestion containing 1 µg of isolated RNA from liver was used for transcription into cDNA by using iScript™ cDNA Synthesis Kit (1708891, Bio-Rad, Sweden) performed according to manufacturer's protocol.

### Quantitative-PCR
The Applied Biosystems™ TaqMan™ Pre-Developed Assay Reagents from Thermo Fisher were used for real-time relative quantitative evaluation of mouse *Stxbp1* gene expression. Experiments were performed according to manufacturer's protocol. Briefly, the anti-mouse *Stxbp1* probe (Mm00435837_m1) and anti-mouse GAPDH probe (Glycerinaldehyd-3-phosphat-Dehydrogenase; 4352932E) purchased from Applied Biosystems™ (by Life Technology) were diluted 1:20 and used to detect murine *Stxbp1* gene segments. GAPDH expression served as reference. All samples were run in duplicates using a Quantstudio-7 machine from applied Biosystems (Thermo Fisher).

### Stxbp1 protein inhibition in RAW-blue™ cells
Cells were seeded in 96 plates at 50,000 cells/ml before starting the experiment. Then, cells were incubated with HiPerFect Transfection Reagent (Qiagen; Cat.NO: 301704) plus either negative siRNA or *Stxbp1* siRNA. After a 48 h incubation period, cells were stimulated overnight with EVs and cell supernatants were used for the NF-κB activation assay.

### In vivo animal models
Animals were housed under standard conditions of light and temperature and had free access to chow and water. For experiments, EVs were collected (Per mouse, $200 \times 10^6$ cells EVs were injected), washed in PBS once and resuspended in cell culture medium. BALB/c mice (female, 8-9 weeks; Janvier Labs) were injected *i.t.* with 100 µl of indicated stimulant suspension (*n* = 6 mice per group). BALB/c mice were also injected intratracheal (*i. t.*) with 50 µl of Y-2763 (10 mg/kg) suspension. 1 h after Y-2763 injection, 50 µl of EVs were administrated *i. t.* PBS, Y-2763, TNFα, IgA2 and heparin injection alone were used as controls (*n* = 6/group). 4 h after post-infection, mice were deeply anesthetized by isoflurane, and the mice were sacrificed to evaluate cytokine levels in blood as well as for SEM of lungs. Blood was immediately collected in 0.1 M sodium citrate (1:10 citrate:blood) by cardiac puncture and lungs were prepared for microscopical analysis. Blood was centrifuged at $1000 \times g$ for 10 min and supernatant was used for analysis of inflammatory mediators as described above.

### Induction of CLP
Abdominal sepsis was induced by puncture of the cecum was employed[45]. For these experiments STXBP1 + /- (Male, 8-9 weeks C57BL/6) mice which have a Stxbp1 expression that is approximately 25% reduced compared to wild-type mice[46]. In brief, the abdomen was opened, and the exposed cecum was filled with feces by milking stool backward from the ascending colon. A ligature was placed below the ileocecal valve. The cecum was soaked with phosphate- buffered saline (PBS; pH 7.4) and punctured twice with a 21-gauge needle. This cecal ligation and puncture (CLP) protocol is associated with less than 10% mortality within 24 h. The cecum was then pushed back into the abdominal cavity and the abdominal incision was sutured.

### Liposomal Stxbp1 RNAi delivery in mouse model
At a dosage of 4 mg/kg, female and 6-9 weeks old BALB/c mice were given siRNA treatment. Silencer Select Predesigned *Stxbp1* siRNA (Thermo Fisher Scientific, 4457308 #s74551) was injected into the tail vein (200 µl) and intratracheal (*i. t.* 100 µl) with Invivofectamine™ 3.0 reagent (Thermo Fisher Scientific, IVF3005). *Stxbp1* siRNA was injected 2 and 1 day before EV challenge (Per mouse, $200 \times 10^6$ cells EVs were injected). As a control, Ambion™ In Vivo Negative Control #1 siRNA was employed (Thermo Fisher Scientific, 4457289). Total RNA and proteins were extracted from the Lungs of *Stxbp1* siRNA–treated animals and control mice for *Stxbp1* gene and protein expression.

### Immunohistochemistry
The lung tissues were harvested from mice and were submerged in 4% buffered formaldehyde, Histofix (Histolab, Göteborg, Sweden). This was followed by a dehydration step and paraffin embedding, 4 µm thin parallel sections were generated from the tissue blocks. After rehydration and antigen retrieval using EnVision Flex, High pH (Agilent Dako, U.S.A) sections were labelled with GFP Polyclonal antibody, Alexa Fluor 594 (A-21312, Invitrogen) and CoraLite®488-conjugated ZO-1 polyclonal antibody (CL488-21773, Proteintech). Light microscopy fluorescent images were visualized on LRi Olympus BX43 microscope. Images were acquired with cellSens standard, version 1.9, (Olympus corporation). For the confocal images the samples were visualized on a Nikon A1+ confocal microscope with a 20 × Plan Apo objective (NA 0.75) or a 60 × Apo DIC oil immersion objective (NA 1.40) (Nikon Instruments Inc.). Images were acquired and processed with NIS-elements, version: 4.60.02, (Laboratory Imaging, Nikon, Tokyo, Japan).

### Preparation of large EVs from human plasma or from cell supernatants
Human plasma samples were prepared by centrifuging blood at $1000 \times g$ for 10 min. Samples were frozen ($-80$ °C) until use. After thawing at 37 °C remaining cells were removed by centrifugation at $300 \times g$ for 10 min. The collected supernatants were subjected to another centrifugation step at $5000 \times g$ for 10 min in order to remove apoptotic bodies and cell debris. EVs were then extracted from the

plasma and cell supernatants by a third centrifugation step at 21,000 × g for 30 min at 20 °C. Control EVs were isolated from citrate plasma from a group of healthy people.

## Protein Digestion and Tandem Mass Tag (TMT) Labeling

Ten μg of each of 8 samples, 3.4 μg from a control sample and a reference pool made from 5 mg from 4 samples (totally 20 μg) were lysed by shaking in in 2% sodium dodecyl sulfate (SDS) in 50 mM triethylammonium bicarbonate (TEAB). The samples were digested with trypsin using the filter-aided sample preparation (FASP) method[47]. Briefly, the samples were reduced with 100 mM dithiothreitol at 60 °C for 30 min. The reduced samples were transferred to 30 kDa MWCO Pall Nanosep centrifugation filters (Pall Corporation), washed several times with 8 M urea and once with digestion buffer (DB, 0.5% sodium deoxycholate in 50 mM TEAB) prior to alkylation with 10 mM methyl methanethiosulfonate in digestion buffer for 20 min in room temperature. Digestions were performed by addition of Pierce MS grade Trypsin (Thermo Fisher Scientific) in DB to a trypsin:protein ratio of 1:100 and incubated overnight at 37 °C. Next morning an additional portion of trypsin was added and incubated for another 3 h. Peptides were collected by centrifugation and labelled using TMT 10-plex isobaric mass tagging reagents (Thermo Scientific) according to the manufacturer's instructions. Labelled samples from each sample type were combined into 2 sets and sodium deoxycholate was removed by acidification with 10% TFA. The combined TMT-labeled samples were desalted using Pierce Peptide Desalting Spin Columns (Thermo Scientific) following the manufacturer's instructions.

## Fractionation and nLC-MS/MS Analysis

Each set was pre-fractionated on the Dionex Ultimate 3000 UPLC system (Thermo Fischer Scientific) using the Waters XBridge BEH C18 column (3.0 × 150 mm, 3.5 μm, Waters Corporation, Milford, USA) and the gradient from 3% to 40% solvent B over 18 min, from 40% to 100% B over 5 min, 100% B for 5 min, all at the flowrate of 0.4 ml/min; solvent A was 10 mM ammonium formate in water at pH 10.0, solvent B was 90% acetonitrile, 10% 10 mM ammonium formate in water at pH 10.0. The 40 primary fractions were concatenated into 20 fractions, evaporated and reconstituted in 3% acetonitrile, 0.2% formic acid for nLC-MS/MS analysis.

Each fraction was analyzed on Orbitrap Fusion Tribrid mass spectrometer interfaced with Easy-nLC 1200 nanoflow liquid chromatography system (Thermo Fisher Scientific). Peptides were trapped on the Acclaim Pepmap 100 C18 trap column (100 μm × 2 cm, particle size 5 μm, Thermo Fischer Scientific) and separated on the in-house packed C18 analytical column (75 μm × 32 cm, particle size 3 μm) using the gradient from 5% to 32% B in 75 min, from 32% to 100% B in 5 min, and 100% B for 10 min at a flow of 300 nl/min. Solvent A was 0.2% formic acid and solvent B was 80% acetonitrile, 0.2% formic acid. MS scans were performed at 120,000 resolution, m/z range 380-1200. MS/MS analysis was performed in a data-dependent mode, with top speed cycle of 3 s for the most intense doubly or multiply charged precursor ions. Precursor ions were isolated in the quadrupole with a 0.7 m/z isolation window, with dynamic exclusion set to 10 ppm and duration of 45 s. Isolated precursor ions were subjected to collision induced dissociation (CID) at 35 collision energy with a maximum injection time of 50 ms. Produced MS2 fragment ions were detected in the ion trap followed by multinotch (simultaneous) isolation of the top 7 most abundant fragment ions for further fragmentation (MS3) by higher-energy collision dissociation (HCD) at 60% and detection in the Orbitrap at 50,000 resolutions, m/z range 100-500.

## Database search and quantification

MS raw data files for the TMT set were merged for relative quantification and identification using Proteome Discoverer version 1.4 (Thermo Fisher Scientific). A database search for each set was performed with the Mascot search engine (Matrix Science) using the Homo Sapiens Swissprot database, version Mars 2017 with 553941sequences. MS peptide tolerance of 5 ppm and MS/MS tolerance for identification of 600 millimass units (mmu), tryptic peptides with zero missed cleavage and variable modifications of methionine oxidation, fixed modifications of cysteine alkylation, N-terminal TMT-label and lysine TMT-label were selected. The detected peptide threshold in the software was set to a significance of FDR 1% by searching against a reversed database and identified proteins were grouped by sharing the same sequences to minimize redundancy. For TMT quantification, the ratios of the TMT reporter ion intensities in HCD MS/MS spectra (m/z 126-131) from raw data sets were used. Ratios were derived by Proteome Discoverer using the following criteria: fragment ion tolerance as 3 mmu for the centroid peak with smallest delta mass and minimum intensity of 2000. Only peptides unique for a given protein were considered for relative quantitation, excluding those common to other isoforms or proteins of the same family. The quantification was normalized using the protein median. Calculations of the ratios were made by using a reference sample made from a mix of 4 of the samples or the control sample as denominator.

The mass spectrometry proteomics raw data have been deposited to the ProteomeXchange Consortium via the PRIDE partner repository with the dataset identifier PXD048039.

## Differentially expressed protein analysis

A Significance Analysis of Microarrays (SAM) using default 500 permutations was performed with TMeV v 4.9 software to identify differentially expressed proteins between polytrauma and sepsis samples[48]. The main benefit of the SAM approach is the computation of the false discovery rate (FDR). The FDR represents the percentage of proteins identified as significant by chance. A low FDR (q-value = 0) for a significant protein would indicate a greater likelihood that the protein represents a true, significant protein, rather than a falsely discovered protein. Heatmap demonstrating the protein expression level changes in plasma samples from polytrauma and sepsis patients are shown. *Red* in the heatmap represents up-regulation, and *Green* represents down-regulation. Gene ontology (GO) enrichment analysis was performed using the Enrichr, a web-based tool providing various types of visualization summaries of collective functions of protein lists[49]. Top 20 enriched GO terms were considered.

## Cell culture

The Human Embryonic Kidney cell line HEK293 (ATCC® CRL-1573™) was purchased from American Type Culture Collection and cultured in DMEM (Gibco® by Thermo Fisher Scientific). HEK-Dual™ TNFα, HEK-Blue™ IL-1β, and HEK-Blue™ IL-10 reporter cells (hkb-TNFα, hkb-il1r, and hkb-il10) were obtained from Invivogen; supplemented with 10% (v/v) heat-inactivated FBS (Invitrogen) and 1% (v/v) antibiotic-antimycotic solution (Invitrogen) at 37 °C in an 5% $CO_2$ incubator. Sub-confluent (70-80%) cells were split by scraping and maintained until passage 17. The human monocyte cell lines THP-1-Dual™ and THP-1-Dual™ KO-MyD cells were purchased from Invivogen, maintained in RPMI Medium 1640 + GlutaMAX™-1 (Gibco® by Thermo Fisher Scientific), 10% (v/v) of heat-inactivated FBS (Invitrogen) and 1% (v/v) antibiotic-antimycotic solution (Invitrogen) at 37 °C in an 5% $CO_2$ incubator. Cells were split every second or third day to a density of $5 \times 10^5$ cells per milliliter. RAW-Blue™ cells were purchased from Invivogen, maintained in DMEM (Gibco® by Thermo Fisher Scientific), 10% (v/v) of heat-inactivated FBS (Invitrogen) and 1% (v/v) antibiotic-antimycotic solution (Invitrogen) at 37 °C in an 5% $CO_2$ incubator. RAW-Blue, HEK293 and THP-1 reporter cells express an NF-κB- and AP1-inducible secreted embryonic alkaline phosphatase (SEAP) reporter gene that is detectable and measurable when using Quanti-Blue™ (Invivogen). For maintain gene selection pressure, recommended antibiotics were used according to the manufacturer's instructions.

Cells were tested negative for mycoplasma but were otherwise not further authenticated.

## NF-κB pathway array

Human Proteome profiler NF-κB Pathway Array Kit was purchased from R&D Systems (Minneapolis, MN, USA) and experiments were performed according to the manufacturer's protocols. THP-1 reporter cells were stimulated with 10 ng/ml of *E. coli* LPS overnight at 37 °C and 5% $CO_2$ and the resulting cell and EV lysates (200 μg) was used for the Proteome profiler Human NF-κB Pathway Array Kit.

## Cell stimulation and NF-κB activation assay

THP-1 and HEK293 reporter cells were stimulated with 10 ng/ml of *E. coli* LPS, 20/ml TNFα, 10 ng/ml IL-1β, and 10 ng/ml IL-10 overnight at 37 °C and 5% $CO_2$ for EVs isolation. THP-1 reporter cells ($2 \times 10^5$ per reaction) were cultured in 96 well plates with RPMI containing 10% heat inactivated FBS. HEK293 reporter cells ($5 \times 10^4$ per reaction) were cultured in 96 well plates with DMEM containing 10% heat inactivated FBS. Cells were incubated with ligands and EVs either in presence or absence of cytochalasin D, heparin, anti-ligand antibodies (TNFα, IL-1β, and IL-10), or an isotypic IgA2 control overnight at 37 °C and 5% $CO_2$. To investigate the influence of Y-27632 in modulating cell-EV fusion, RAW1-xBlue™ cells were incubated with 10 μM Y-27632 for 1 h at 37 °C in an 5 % $CO_2$ incubator and later stimulated over night with EVs, isolated from RAW1-xBlue™ cells upon LPS stimulation 1 μg/ml. The pellets were resuspended in 100 μl 1 × SDS-loading buffer. After incubation, cells were spun down at $300 \times g$ for 5 min and 20 μl of the supernatants were used for the NF-κB activation assay. Further adding 180 μl of Quanti-Blue™ (Invivogen) the supernatants were incubated at 37 °C. The absorbance was measured at 600 nm before negative controls begin to be colored. Cells in the absence of stimulant served as negative control whereas cells incubated with 10 ng/ml LPS, 20 ng/ml TNFα, 10 ng/ml IL-1β, and 10 ng/ml IL-10 represented positive controls.

## EV trypsinization

Freshly isolated EVs were trypsinised with Trypsin-EDTA (100 μg/ml) at 37 °C for 10 min (Cat#25200056; Gibco™, Thermo Fisher) and were washed before adding to the cells. EVs were then pelleted and resuspended in cell culture medium. Undigested EVs were incubated with PBS.

## Cytokine luminex assay

The levels of 45 human cytokines/chemokines/growth factors were measured in supernatants of whole blood stimulation using the multiplex Immunoassay ProcartaPlex® (EPX45.12171-901; affymetrix eBioscience/Thermo Fisher Scientific, Bender MedSystems GmbH) according to the manufacturer's instructions. Murine cytokines and chemokines were measured in citrate plasma of infected mice by Multiplex Immunoassay ProcartaPlex® mCytokine/Chemokine Panel 1 A 36plex (EPX36.26092-901; affymetrix eBioscience, Bender Med-Systems GmbH) or by using Mouse Cytokine customized 8-Plex performed according to the manufacturer's instructions and analyzed in a flow-based Bio-Plex™ 200 system (Bio-Rad).

## Bradford assay

To determine the cells and EV protein concentration the Pierce 660 nm Protein Assay Reagent was employed using standard series provided in the kit. The standards contain bovine serum albumin (BSA) in different concentrations from 125 to 2000 μg/μL. As a blank distilled water was used. Samples were diluted until it fitted into the standard series. For measurements 150 μl Assay Reagent was added to 10 μl diluted sample, blank or standard. The photometric measurement was performed directly after adding of the Assay Reagent at 660 nm.

## Co-immunoprecipitation

EVs were separated from the cells, lysed in IP buffer (cat#87788; Thermo Fisher) with Protease and Phosphatase Inhibitor (Cat#1861281; Thermo Fisher), and agitated for 1 h at 4 °C shaking. The IP buffer consists of 1% NP40, 125 mM NaCl and 25 mM Tris/HCL pH 7.4. Supernatants were transferred to a new reaction tube and a Bradford assay performed. The samples were stored at −80 °C until use. Lysed samples were preincubated with beads for 30 min at 4 °C and centrifuged to collected supernatants. Mouse anti-IL1R1 antibody (Cat# sc-393998; Santa Cruz) was then added to the protein sample. The antibody-protein complex was incubated overnight at 4 °C with agitation. Prewashed beads were added to the sample and incubated for 3 h by shaking at 4 °C. Afterwards, the magnetic beads were separated from the supernatants and transferred to a new tube where they were washed trice. Finally, samples were resuspended in 2 × reducing Laemmli SDS sample buffer. Unbound samples served as a control. After an incubation for 10 min at 70 °C the beads were removed and the supernatants of the elutions were stored at -20 °C. Rabbit anti-pIRAK4 (Cat#11927; Cell signaling Technology®) and rabbit anti-MyD88 (Cat#4283; Cell signaling Technology®) antibodies were used for WB detection.

## Stimulation of THP1 and RAW cells

For 0 (unstimulated), 2, 4, 6, 8 and ON, 100 ng/ml LPS and 100 ng/ml LTA were used to stimulate THP1 cells, whereas 1000 ng/ml LPS and 100 ng/ml LTA were used to stimulate RAW cells. Later, cells lysates were separated on an SDS-PAGE, transferred, and detected for STXBP1 (ab3451; Abcam) and GAPDH (10494-1-AP; Proteintech). See also Supplementary Fig. 2.

## SDS-PAGE and western blot

Denaturated samples were separated on 10% Criterion™ precast gels in 1×Tris-Glycine-SDS running buffer (Bio-Rad) for 60 min at 100 V. For detecting phosphorylated proteins, fluorescent staining of SDS-polyacrylamide gels (Cat#4561034; Bio-Rad) using Pro-Q Dia-mond phosphoprotein gel stain (Cat#P33300; Thermo Fisher) was performed by fixing the gels in 45% methanol, 5% acetic acid overnight. This was followed by a washing step with three changes of deionized water for 10 to 20 min per wash, followed by incubation in Pro-Q Diamond phosphoprotein gelstain for 180 min, and destaining with successive washes of 15% 1,2-propanediol or 4% acetonitrile in 50 mM sodium acetate, pH 4.0. Following image acquisition, gels were stained with GelCode™ Blue Safe Protein Stain (Thermo Fisher Scientific) and documented using a GelDoc™ EZ Imager.

For Western blot analysis, proteins were transferred onto PVDF-membrane (Immobilon®-P Membrane, EMD Millipore Corporation, USA), blocked with 3% non-fat dry milk or 3% BSA in phosphate-buffered saline (PBS) containing 0.05% Tween-20 for 60 min at RT and incubated with monoclonal rabbit anti-human pIRAK4, (Cat#11927; Cell signaling Technology®), pTRAF2 antibodies (Cat#13908; Cell sig-naling Technology®), anti-human MyD88 (Cat#4283; Cell signaling Technology®), and TRADD antibodies (Cat# sc-46653; Santa Cruz), anti-Munc 18-1 or STXBP1 (Cat#ab3451; Abcam) diluted 1:1000 in blocking solution for 60 min at RT. After a washing step, blots were incubated with secondary antibodies horseradish peroxidase-conjugated goat anti-rabbit IgG (Cat#172-1019; Bio-Rad; 1:1000 in blocking solution) for 45 min at RT. Membranes were washed with PBS-Tween and developed with Super Signal® West Pico Chemiluminescent Substrate (Cat#34078; Thermo Scientific).

## Transfection of HEK293 cells

HEK293 cells were transfected with the TNFRSF1A (Cat.no: HG10872-ACG, Sinobiological) plasmid DNA using calcium phosphate transfec-tion protocol (ProFection Mammalian Transfection System-Calcium Phosphate (REF-E1200, Promega Corporation, USA). 24 h after the

transfection 200 µg/ml of Hygromycin B was added to the cell culture medium for selective propagation of the transfected cells. The transfected cells have GFP encoded in the plasmid. Once the cells were about 90% confluent, they were resuspended in PBS with 2% serum to a cell density of $5 \cdot 10 \times 10^6$ cells/ml. The sorting was done on an Ariallu SORP instrument from BD Biosciences. The cells were excited by a 100 mW 488 nm laser and the GFP expression was detected through a 530/30 bandpass filter. The cells were single cell sorted through a 100 µm nozzle into 96 well plates.

## Scanning electron microscopy
For scanning electron microscopy (SEM), cells/EV were fixed overnight at RT with 2.5% glutaraldehyde in 0.15 M cacodylate buffer, pH 7.4, followed by four washing steps with cacodylate buffer for 10 min and dehydration by an ascending ethanol series from 50% (v/v) to absolute ethanol. Samples were subjected to critical point drying with carbon dioxide ($3 \times 10$ min) using critical point dryer CPD 030 (Bal-Tec). The tissue/cell samples were mounted on aluminum holders ($32 \times 5$ mm) with 25 nm carbon tabs, silver paint and sputtered with 20 nm palladium/gold by using vacuum coater Leica EM ACE200 (Leica Microsystems A/S). Samples were examined in a PHENOM PROX scanning electron microscope. Reagents and equipment were purchased from Agar scientific.

## Statistical analysis
In vitro experiments were done in triplicates in two independent trials. Data were analyzed with Microsoft Excel 2021 (Microsoft, Redmond, WA) and GraphPad Prism 9.0 (GraphPad Software, San Diego, CA). All results are presented as mean values ± SEM with the number of independent experiments and mice per group indicated in the figure legends. Comparison of data was performed by one-way or two-way ANOVA and Tukey's multiple comparison test.

## Reporting summary
Further information on research design is available in the Nature Portfolio Reporting Summary linked to this article.

## Data availability
All proteomics data in the study have been made public. The mass spectrometry proteomics raw data have been deposited to the ProteomeXchange Consortium via the PRIDE partner repository with the dataset identifier PXD048039. Source data used in the main figure panels are also provided. Source data are provided with this paper.

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

## Acknowledgements

This work was supported in part by the Alfred Österlund Foundation (to P.P. and H.H.), the Craford Foundation (grant number 20180506 and 20210908 to P.P.), the Medical Faculty at Lund University (to H.H.), the Swedish Foundation for Strategic Research (grant number SB12-0019 to H.H.), the Swedish Research Council (grant number 2013-3078 to H.H.), the Stig and Ragna Gorthon Foundation, Helsingborg to C.R. and J.O.). The funders had no role in study design, data collection and analysis, decision to publish, or preparation of the manuscript. The Proteomics Core Facility at Sahlgrenska Academy, Gothenburg University, performed the analysis for protein identification. We are grateful of Inga-Britt and Arne Lundbergs Forskningsstiftlese for the donation of the Orbitrap Fusion Tribrid MS instrument. We would like to thank Matthijs Verhage and Joke Wortel (University Amsterdam, The Netherlands) for donating *Stxbp1* KO mice. We would like to thank Tobias Schmidt at the Reumatologi Facility Lund University. Additionally, we would like to thank Dr. Lloyd Tanner for his assistance in editing the manuscript.

## Author contributions

P.P., I.T., K.K., E.B., R.B., E.S., M.B., E.C., L.G., and A.N. performed the in vitro and in vivo experiments. M.R performed CLP model and H.T analyzed the data. S.V. generated heat maps. J.O., C.R., and M.H. assisted when performing experiments with blood from patient samples. P.P and H.H. designed, analyzed data, and supervised the study, and wrote the manuscript.

## Funding

## Competing interests

The authors declare no competing interests.

## Additional information

**Peer review information** : *Nature Communications* thanks Rama Gurram and the other, anonymous, reviewer(s) for their contribution to the peer review of this work. A peer review file is available.

