## [Peer Review File · Nature Communications]

The Role of Extracellular Vesicle Fusion with Target Cells in Triggering Systemic InflammationEditorial Note: This manuscript has been previously reviewed at another journal that is not operating a transparent peer review scheme. This document only contains reviewer comments and rebuttal letters for versions considered at *Nature Communications*.

REVIEWER COMMENTS

Reviewer #1 (Remarks to the Author):

The authors have only partially addressed my concerns and my concerns regarding novelty remain. I have no further comments to add.

Reviewer #2 (Remarks to the Author):

Although the manuscript's current version is mostly similar to the previous version, the author's response is satisfactory. The current version still holds the interesting part: EVs can exert inflammatory activity in cells that do not respond to inflammatory stimuli. The current version is satisfactory to me, and I hope it will also interest the readers. I want to suggest to the editor that the current version of the manuscript may be accepted for publication.

Reviewer #4 (Remarks to the Author):

The Authors have addressed the concerns by adding new experimental data and through textual revisions. Together, the revisions strengthen the manuscript that introduces interesting new findings to appreciate inflammatory signaling properties conferred by EVs.

Reviewer #5 (Remarks to the Author):

I have now reviewed the manuscript mainly limited to the response of the authors to Reviewer #3. I find that overall the authors response satisfied the demands and suggestions made by Reviewer #3 with one exception, see below:

Major concern

1. In Figure 1, the authors primarily used flow cytometry to detect specific marker expression to demonstrate that neutrophils or monocytes can absorb platelet-derived EVs. Can this be further validated by more direct tracing methods? Additionally, it is unclear why the authors shifted their focus from analyzing platelet-derived EVs to THP1 cell-derived EV components. What are the differences between EVs from these two sources, and which one is the predominant mode in inflammation? The authors need to clarify these points.

Response:

Apart from flow cytometry, also nanoparticle tracking analysis and electron microscopy can be used to characterize EVs¹. Unfortunately, we have no access to nanoparticle tracking analysis. Therefore, we tried to visualize EVs by electron microscopy, however, as plasma is a complex fluid, the results were not conclusive and therefore we not to proceed with an electron microscopical analysis.

Reviewer #4: I think there was a misunderstanding regarding the reviewers request. The authors show that after LPS stimulation, a marker associated with platelet-derived EVs, CD42b, is increased on the surface of neutrophils and monocytes. The possibility of an endogenous up regulation of CD42b has to be excluded or at least diminished. Here it would be important to show uptake of EVs from plasma by, for example, prior fluorescent labeling and then microscopic images of recipient cells with the fluorescent marker as well as an increase in fluorescence uptake after LPS stimulation (the latter could be done again by FACS).

Though we focussed on monocyte-derived EVs, we studied platelet-derived EVs in the first series of experiments because we aimed to demonstrate the physiological relevance of our fusion concept. Notably, platelets are by far the most abundant releaser of EVs which would have made it extremely difficult to study monocyte-derived EVs in a such complex fluid such as plasma. Once we were able to prove our concept we switched to a more clean system as plasma that allowed us to study monocyte-derived EVs. As these EVs are released from immunomodulatory cells, we felt that these cells are from a pathophysiological point of view of more relevance.

Reviewer #4: I agree that monocyte-derived EVs are important in the inflammatory reaction and are therefore relevant and sufficient for further experiments.

...

Minor concern

1. In Figure 1, it is necessary to verify the purity of EVs. It is recommended to verify the purity from the perspective of key biomarker expression and vesicle size.

Response:

The purity of EVs is shown in supplemental Figure 3 (right panel).

Reviewer #4: I'm afraid I cannot find this Figure. Supplemental Figure 3 in my files shows quantification of the release of cytokines and I can't find any other potential data such as WB with EV specific markers.

We appreciate the feedback provided by the reviewers, which we will use to improve our study. In response to the reviewers' comments, we will address the concerns raised by the reviewers as outlined point by point below.

REVIEWER COMMENTS

Reviewer #1 (Remarks to the Author):

The authors have only partially addressed my concerns and my concerns regarding novelty remain. I have no further comments to add.

***Response:** As detailed in our previous response, we have already incorporated additional information to clarify the novelty of our work. This includes elucidating the mechanisms rendering unresponsive cells susceptible to an otherwise non-stimulating inflammatory ligand, uncovering the molecular processes by which heparin down-regulates extracellular vesicle-mediated inflammation, and providing insights into why neutralizing antibodies fail to block extracellular vesicle-triggered signaling. Furthermore, we also discovered that the transport of intracellular signaling adaptor molecules, such as MyD88, IRAK4, TRAF2, and TRADD, to their target cells, which to our knowledge has not yet been reported by others. This has now been specially highlighted in the revised manuscript (lines 460-463).*

Reviewer #2 (Remarks to the Author):

Although the manuscript's current version is mostly similar to the previous version, the author's response is satisfactory. The current version still holds the interesting part: EVs can exert inflammatory activity in cells that do not respond to inflammatory stimuli. The current version is satisfactory to me, and I hope it will also interest the readers. I want to suggest to the editor that the current version of the manuscript may be accepted for publication.

***Response:** Thank you for your positive feedback on the manuscript. We appreciate the reviewer's overall conclusion and are pleased that the current version aligns with your expectations.*

Reviewer #4 (Remarks to the Author):

The Authors have addressed the concerns by adding new experimental data and through textual revisions. Together, the revisions strengthen the manuscript that introduces interesting new findings to appreciate inflammatory signaling properties conferred by EVs.

***Response:** We were pleased to learn that the reviewer acknowledged the new experiments and suggested accepting the manuscript.*

Reviewer #5 (Remarks to the Author):

I have now reviewed the manuscript mainly limited to the response of the authors to Reviewer #3. I find that overall the authors response satisfied the demands and suggestions made by Reviewer #3 with one exception, see below:

Response: We were glad to read that the reviewer was satisfied with the revised version of our manuscript.

Major concern

1. In Figure 1, the authors primarily used flow cytometry to detect specific marker expression to demonstrate that neutrophils or monocytes can absorb platelet-derived EVs. Can this be further validated by more direct tracing methods? Additionally, it is unclear why the authors shifted their focus from analyzing platelet-derived EVs to THP1 cell-derived EV components. What are the differences between EVs from these two sources, and which one is the predominant mode in inflammation? The authors need to clarify these points.

Response:

Apart from flow cytometry, also nanoparticle tracking analysis and electron microscopy can be used to characterize EVs¹. Unfortunately, we have no access to nanoparticle tracking analysis. Therefore, we tried to visualize EVs by electron microscopy, however, as plasma is a complex fluid, the results were not conclusive and therefore we not to proceed with an electron microscopical analysis.

Reviewer #4: I think there was a misunderstanding regarding the reviewers request. The authors show that after LPS stimulation, a marker associated with platelet-derived EVs, CD42b, is increased on the surface of neutrophils and monocytes. The possibility of an endogenous up regulation of CD42b has to be excluded or at least diminished. Here it would be important to show uptake of EVs from plasma by, for example, prior fluorescent labeling and then microscopic images of recipient cells with the fluorescent marker as well as an increase in fluorescence uptake after LPS stimulation (the latter could be done again by FACS).

Though we focussed on monocyte-derived EVs, we studied platelet-derived EVs in the first series of experiments because we aimed to demonstrate the physiological relevance of our fusion concept. Notably, platelets are by far the most abundant releaser of EVs which would have made it extremely difficult to study monocyte-derived EVs in a such complex fluid such as plasma. Once we were able to prove our concept we switched to a more clean system as plasma that allowed us to study monocyte-derived EVs. As these EVs are released from immunomodulatory cells, we felt that these cells are from a pathophysiological point of view of more relevance.

Reviewer #4: I agree that monocyte-derived EVs are important in the inflammatory reaction and are therefore relevant and sufficient for further experiments.

Response: Thank you for your positive feedback and agreement regarding the significance of monocyte-derived EVs in the inflammatory reaction which is greatly appreciated. We will address the concerns raised by the reviewer as outlined below.

Minor concern

1. In Figure 1, it is necessary to verify the purity of EVs. It is recommended to verify the purity from the perspective of key biomarker expression and vesicle size.

Response (major & minor comments): The experiment proposed by the reviewer, i.e., demonstrating the uptake of EVs from plasma using fluorescent labeling and providing

microscopic images of recipient cells with the fluorescent marker, along with an increase in fluorescence uptake after LPS stimulation (measurable by FACS), EVs size and purity has already been published by Chimen et al. in 2020¹. This is for instance shown in Figure 2C of their article depicting representative images of monocytes labeled with anti-CD14, anti-GPIIb/IIIa (note that GPIIb/IIIa and CD42b represent the same protein), platelet-derived extracellular vesicles (PEVs), and DAPI captured through confocal microscopy. We have now included this information in the revised version of our manuscript (line 119).

Regarding the second question asked by the reviewer about the upregulation of CD42b on neutrophils and monocytes, it is important to emphasize that the CD42b antibody serves as a commercially available platelet biomarker. In addition, the Protein Atlas states that only platelets and megakaryocytes express CD42b, while no other cells are mentioned (www.proteinatlas.org/ENSG00000185245-GPI1BA). Thus, we are very confident that neutrophils and monocytes cannot up-regulate CD42b on their surface upon stimulation. This was also confirmed in a literature search where we were not able to find an article that describes endogenously expressed CD42b by neutrophils and monocytes. To address the reviewer's concern experimentally, we conducted FACS analysis in which purified neutrophils were either stimulated with LPS or left untreated. At various time points, CD42b expression was measured using a murine CD42b antibody. Under both experimental conditions, we did not observe an endogenous upregulation of CD42b, as depicted in Figure 1. This information has been added to the revised version of our manuscript (lines 742-746).

In a second experiment, we incubated human blood with fluorescently-labeled antibodies against CD42b and CD66b (a neutrophil marker). Following a 60-minute incubation, unbound antibodies were removed by a centrifugation step. The cells were washed and then resuspended in a PBS buffer containing 0.5% BSA and either stimulated with LPS or left untreated. Subsequent FACS analysis was conducted to visualize the fusion of platelet-derived extracellular vesicles. As demonstrated in Figure 2 (see below), when blood was exposed to LPS, an increased number of CD42b-positive neutrophils was observed which was greatly reduced in the control experiment. Also this experiment confirms our conclusion that platelet-derived CD42b-positive extracellular vesicles are able to fuse with their target cells and that the results we obtained are not caused by endogenously expressed CD42b. Both sets of results are now mentioned in the manuscript as data not shown.

Reference:

- 1 Chimen, M. et al. Appropriation of GPIIb/IIIa from platelet-derived extracellular vesicles supports monocyte recruitment in systemic inflammation. *Haematologica* **105**, 1248-1261 (2020). <https://doi.org/10.3324/haematol.2018.215145>

Figure 2: Flow cytometry analysis of human blood upon LPS-challenge are presented. The cell fractions that were examined are labelled. The translocation efficiency of CD42b, a platelet activation marker, to Neutrophils from healthy (right panel) and LPS-treated blood is depicted (right panel). The figure shows that extracellular vesicles that are already in complex with an antibody against CD42b fuse with neutrophils.

Material and Methods:

Blood from healthy donors was collected in citrate tubes (BD). Blood was mixed 1:1 with 1 x PBS (Gibco) and layered on top of Ficoll Paque Plus (Millipore Sigma). Tubes were spun for 30 min at 400 g. Plasma was removed, the PBMC fraction was transferred to a new tube, and washed with 1 x PBS. The pellet resulting from the gradient centrifugation contains erythrocytes and granulocytes. Red blood cells were lysed with RBC buffer twice (see for reference <https://www.ncbi.nlm.nih.gov/pmc/articles/PMC8473578/pdf/main.pdf>) and the pellets were washed with 1 x PBS.

Flow cytometry:

Blood and isolated neutrophils from two healthy donors were incubated with CD66b-FITC (Clone: G10F5, BD Bioscience) and CD42b-PE-CyTM5 (Clone: HIP1, BD Bioscience) antibodies were added (1:50) to the cell suspension and incubated for 60 min at 37 °C in the dark. Samples were washed once with 0.5% BSA in PBS and stimulated with 0.5 µg/ml *E. coli* O111:B4 LPS (Sigma-Aldrich®) for 15 min at 37 °C. Red blood cells were lysed with a BD PhosflowTM Lyse/Fix 5X. After lysis, samples were washed once with 0.5% BSA in PBS and the cell pellet was resuspended in 300 µl of washing buffer. The percentage of CD42b on the cell surface was calibrated with control cells that were not treated with LPS.

Response:

The purity of EVs is shown in supplemental Figure 3 (right panel).

Reviewer #4: I'm afraid I cannot find this Figure. Supplemental Figure 3 in my files shows

quantification of the release of cytokines and I can't find any other potential data such as WB with EV specific markers.

***Response:** We apologize for stating the wrong figure number, the purity of EVs is shown in supplemental Figure 4 (right panel).*

REVIEWERS' COMMENTS

Reviewer #4 (Remarks to the Author):

The authors have satisfactorily responded to the requested revision. I have no further comments.